# Expression noise facilitates the evolution of gene regulation

**Luise Wolf[1,2†], Olin K Silander[1,2*‡], Erik van Nimwegen[1,2*]**

[1]Biozentrum, University of Basel, Basel, Switzerland; [2]Swiss Institute of Bioinformatics, Basel, Switzerland

**Abstract** Although it is often tacitly assumed that gene regulatory interactions are finely tuned, how accurate gene regulation could evolve from a state without regulation is unclear. Moreover, gene expression noise would seem to impede the evolution of accurate gene regulation, and previous investigations have provided circumstantial evidence that natural selection has acted to lower noise levels. By evolving synthetic *Escherichia coli* promoters de novo, we here show that, contrary to expectations, promoters exhibit low noise by default. Instead, selection must have acted to increase the noise levels of highly regulated *E. coli* promoters. We present a general theory of the interplay between gene expression noise and gene regulation that explains these observations. The theory shows that propagation of expression noise from regulators to their targets is not an unwanted side-effect of regulation, but rather acts as a rudimentary form of regulation that facilitates the evolution of more accurate regulation.

**\*For correspondence:**
olinsilander@gmail.com (OKS);
erik.vannimwegen@unibas.ch
(EN)

**Present address:** [†]Roche Pharma Research and Early Development, Roche Innovation Center Basel, F. Hoffmann-La Roche Ltd, Basel, Switzerland; [‡]Institute of Natural and Mathematical Sciences, Auckland Campus, Massey University, Auckland, New Zealand

**Competing interests:** The authors declare that no competing interests exist.

## Introduction

Many studies over the last decade have established that, even within homogeneous environments, gene expression varies across genetically identical cells due to thermodynamic fluctuations in the molecular events underlying gene expression and the small numbers of molecules involved (*Elowitz et al., 2002*; *Rao et al., 2002*). This phenomenon is commonly referred to as 'expression noise' (*Blake et al., 2003*; *Raser and O'Shea, 2005*). Much progress has been gained in understanding the molecular mechanisms underlying noise in gene expression, and noise in transcription in particular (see, for example, *Sanchez and Golding, 2013*). In the simplest scenario, basic thermodynamic fluctuations and Brownian motion of the molecular players would cause transcription initiation at a given promoter to occur with a constant probability per unit time, and the corresponding mRNAs to decay with a constant probability per unit time, leading to a Poissonian steady-state distribution in the number of transcripts. Although such Poissonian fluctuations are observed for some genes, most genes exhibit much larger fluctuations in their mRNA copy number. It is generally believed that such increased variability originates in promoters stochastically switching between different states that are associated with different transcription initiation rates. In the simplest scenario, promoters stochastically switch between an 'on' and 'off' state, producing 'bursts' of transcript while in the on state, and this would lead to increased noise as has been well understood theoretically (*Kepler and Elston, 2001*; *Paulsson, 2005*; *Shahrezaei and Swain, 2008*). However, events such as the stochastic binding and unbinding of transcription factors (TFs), or modifications of the local chromatin state, would generally cause most promoters to switch between a much larger number of different states. Moreover, the extent of promoter state switching would be expected to depend on the specific promoter architecture. Indeed, several studies have shown that different promoters show different amounts of gene expression noise, and that these differences are, at least to some extent, encoded in the promoter sequence (*Newman et al., 2006*; *Hornung et al., 2012*; *Silander et al., 2012*; *Carey et al., 2013*; *Jones, et al., 2014*).

**eLife digest** Genes are stretches of DNA that contain the instructions needed to make proteins and other molecules. By changing how much protein is produced from each gene (i.e., its expression), many organisms—including humans—can produce a wide variety of cell types with very different behaviors. Similarly, single-celled organisms, such as bacteria, can adapt to survive and grow in different environments by changing gene expression levels. It is thus thought that gene expression must be precisely controlled.

However, the molecular processes involved in gene expression are subject to random fluctuations, and so gene expression is inherently 'noisy'. This means that even groups of identical cells in identical environments will show variation in their gene expression patterns. Furthermore, different genes show different levels of noise. The DNA sequence of a part of each gene, called the promoter, has a big effect on these noise levels. Consequently, gene expression noise is a genetically encoded trait, and can therefore be shaped by natural selection. But it remains largely unclear how natural selection has affected gene expression noise.

Now, Wolf et al. have carefully measured the gene expression noise of hundreds of synthetic promoters that were evolved in the laboratory from random DNA sequences, and a similar number of natural promoters in a bacterium called *E. coli*. These experiments revealed that, contrary to expectation, most lab-evolved promoters had low levels of noise. On the other hand, many natural promoters had high levels of noise. Wolf et al. also found that noisy promoters tend to be highly regulated by transcription factors: the proteins that control gene expression by binding to promoter regions. Together, these results imply that unregulated promoters start by having low noise as a default state. Selection pressures must then have caused some *E. coli* promoters to become regulated by transcription factors and raise their noise levels. But, what might these selection pressures have been?

Many genes need to be expressed at different levels in different conditions, and it is generally accepted that regulation by transcription factors evolves to 'satisfy' these requirements. However, transcription factors are themselves noisy, and this noise necessarily propagates to their target genes. Wolf et al. have now developed a general theory showing that this noise-propagation can often benefit an organism. This explains why natural selection can favor an increase in noise levels for regulated genes. Importantly, by showing that the main role of a transcription factor can be to increase the noise of its targets, it suddenly becomes very easy to see how new gene regulatory interactions can evolve from scratch. The next steps in understanding of how gene expression noise evolves will involve manipulating the expression noise of a gene, and measuring how selection acts on such changes.

Importantly, transcriptional noise is thus likely an evolvable trait that is subject to natural selection, but it is currently largely unclear how noise levels have been shaped by natural selection (*Raj and van Oudenaarden, 2008*). On the one hand, it can be argued that in each condition there is an optimal expression level for each protein, such that variations away from this optimal level are detrimental to an organism's fitness, implying that selection will act to minimize noise. Indeed, by investigating the association between expression noise and various statistics that can be considered proxies of organismal fitness, several studies have provided evidence that selection generally acts to minimize noise (*Newman et al., 2006*; *Barkai and Shilo, 2007*; *Lehner, 2010*; *Lehner, 2008*; *Silander et al., 2012*). In this interpretation, genes with lowest noise have been most strongly selected against noise, whereas high noise genes have experienced much weaker selection against noise. On the other hand, gene expression noise generates phenotypic diversity between organisms with identical genotypes, and there are well-established theoretical models showing that such phenotypic diversity can be selected for in fluctuating environments (*Bull, 1987*; *Kussell and Leibler, 2005*). In support of such theoretical models, a number of studies provided examples in which there is a positive association between expression noise and growth (*Blake et al., 2006*; *Bishop et al., 2007*; *Ackermann et al., 2008*; *Zhang et al., 2009*). It is thus possible that some of the genes with elevated noise may have been selected for phenotypic diversity.

# Results

In order to assess how natural selection has acted on the transcriptional noise of promoters, it is critical to determine what default noise levels would be exhibited by promoters that have *not* been selected for their noise properties. To address this, we evolved a large set of synthetic *Escherichia coli* promoters de novo in the laboratory using an experimental protocol in which promoter sequences were selected on the basis of the mean expression level they conferred, while experiencing virtually no selection on their noise properties (*Figure 1*). We synthesized a pool of random DNA sequences, 100–150 base pairs in length, and cloned these upstream of a sequence containing a strong ribosomal binding site and the open reading frame of green fluorescent protein (GFP). Beginning with a library of more than 1 million random promoter clones, we used fluorescence activated cell sorting (FACS) to select cells expressing specific levels of GFP (*Figure 1A–C*). After sorting, we used PCR mutagenesis to input more genetic variation into the library of promoters and repeated the sorting. After the initial FACS sort, this strategy of mutagenesis followed by FACS was repeated four times. The result was a genetically diverse collection of functional promoters that conferred expression close to a pre-specified target level. We selected a subset of 479 synthetic promoters from the third and fifth rounds of FACS selection, choosing equal numbers of promoters from each of six replicate lineages we evolved (*Figure 1*; 'Materials and methods'). We then used flow cytometry, as described previously (*Silander et al., 2012*), to measure the distribution of fluorescence levels per cell for each synthetic promoter, as well as for all native *E. coli* promoters (*Zaslaver et al., 2006*). We used quantitative Western blotting to confirm that the mean fluorescence levels were directly proportional to GFP molecule numbers (*Figure 1—figure supplement 2* and Appendix 1), which allowed us to express fluorescence levels in units of numbers of GFP molecules.

Observing that the fluorescence distributions across cells were well approximated by log-normal distributions (*Figure 1C*), we characterized each promoter's distribution by the mean and variance of log-fluorescence, defining the latter as the promoter's noise level (*Figure 1D*). This definition of noise is equivalent to the square of the coefficient of variation whenever fluctuations are small relative to the mean (Appendix 1), which applies to most promoters, that is, the variance is less than 0.25 for 75% of all promoters (*Figure 1D*).

Although our reporter constructs measure protein levels, that is, GFP, the differences in the noise levels of the reporters are likely dominated by differences in transcriptional noise of the promoters. First, the only differences between the different constructs are the promoter sequence inserts. Consequently, the mRNAs of the different reporters are almost identical, varying only by the short sequence segment between the transcription start site and the constant part of the construct. Second, the reporters were constructed specifically to measure transcription, and feature a constant 5′ UTR part upstream of the start codon of the GFP gene, including a strong ribosomal binding site (*Zaslaver et al., 2006*). Using qPCR we confirmed that protein levels were determined primarily by mRNA levels (*Figure 1—figure supplement 3* and Appendix 1). Because protein decay and dilution rates are identical for all reporters, this implies translation rates vary little across the reporters. Although we have not explicitly measured mRNA decay rates of the reporters, we presume that, because the mRNAs are nearly identical, and because translation rates vary little across the reporters, mRNA decay rates likely vary also only moderately across the reporters. Finally, we note that noise levels were reproducible across biological replicates (*Figure 1—figure supplement 4*), and noise levels estimated using microscopy were consistent with those measured by flow cytometry (*Figure 1—figure supplement 5*).

Importantly, although the differences in noise levels are likely due to differences in transcriptional noise, fluctuations in translation and dilution rates will also contribute to total noise levels that we observe. Indeed, as expected (*Bar-Even et al., 2006*; *Newman et al., 2006*), we observed a systematic relationship between the mean and variance of expression levels of each promoter (*Figure 1D*). In particular, we observed a strict lower bound on variance as a function of mean expression. This lower bound is well described (*Figure 1D*, green curve) by a simple model that incorporates background fluorescence, an intrinsic noise component which is proportional to the number of proteins produced per mRNA, and an extrinsic noise component which likely reflects overall fluctuations in transcription, translation, and dilution rates, that all reporters are subject to (*Taniguchi et al., 2010*) (Appendix 1). We defined the *excess noise* of a promoter as its variance above and beyond this lower bound, allowing us to compare the noise levels of promoters with different means (*Figure 1—figure supplement 6*).

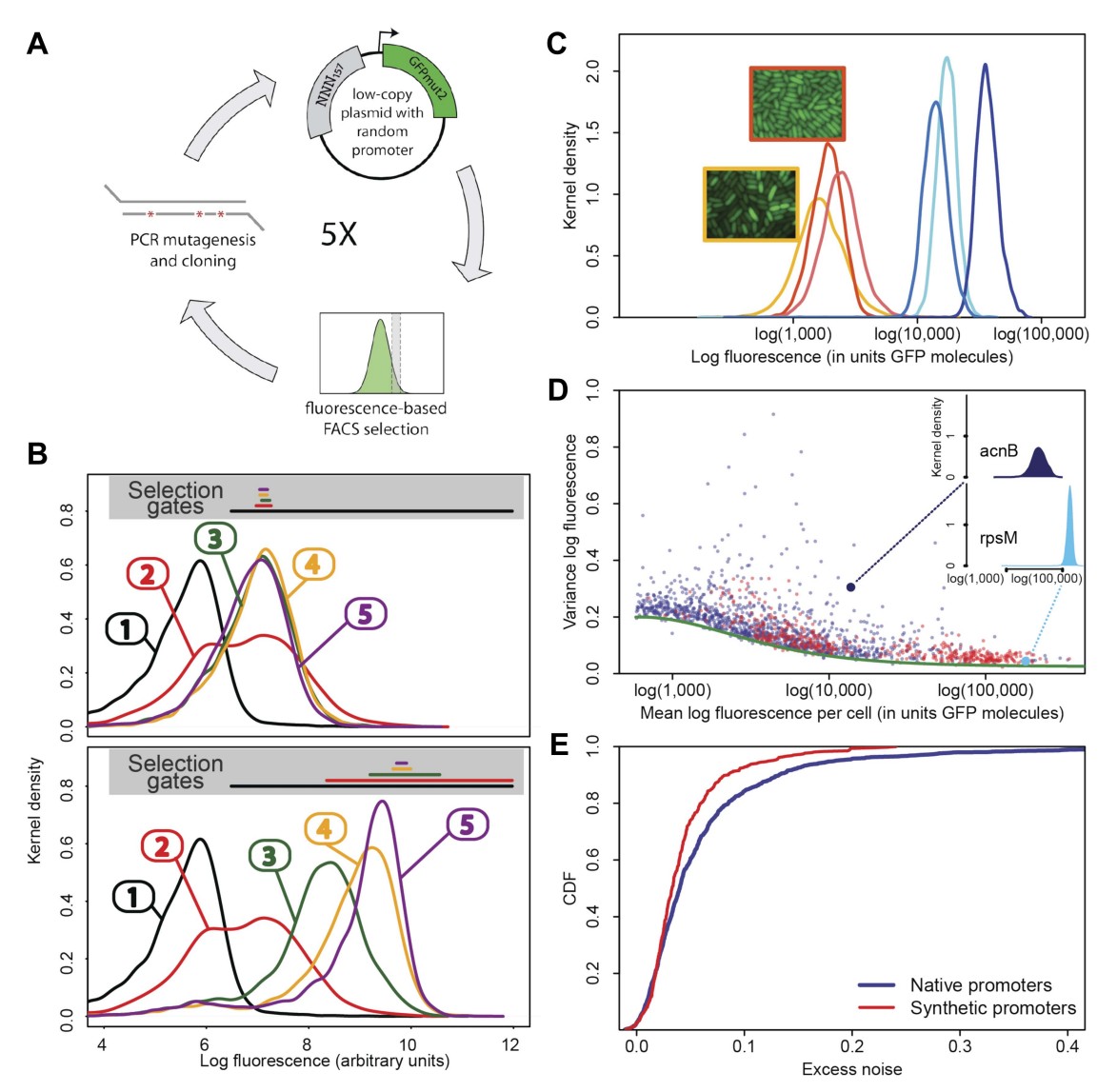

**Figure 1.** Experimental evolution of functional promoters de novo. (**A**) We created an initial library of approximately $10^6$ unique synthetic promoters by cloning random nucleotide sequences, of approximately 100–150 base pairs (bp) in length, upstream of a strong ribosomal binding site followed by an open reading frame for GFP, as used to quantify the expression of native *E. coli* promoters (**Zaslaver et al., 2006**), and transformed this library into a population of cells ('Materials and methods'). We evolved populations of synthetic promoters by performing five rounds of selection and mutation on this library. In each round we used fluorescence activated cell sorting (FACS) to select $2 \times 10^5$ cells that lie within a gate comprising the 5% of the population closest in fluorescence to a given target level. Next, plasmids were isolated from the selected cells and PCR mutagenesis was used to introduce new genetic variation into the promoters. We then re-cloned the mutated promoters into fresh plasmids and transformed them into a fresh population of cells. We performed this evolutionary scheme on three replicate populations in which we selected for a target expression level equal to the median expression level (50th percentile) of all native *E. coli* promoters and three replicate populations in which we selected for a target expression level at the 97.5th percentile of all native promoters (referred to here as medium and high expression levels, respectively). (**B**) Changes in the fluorescence distribution for one evolutionary run selecting for medium target expression (top) and one evolutionary run selecting for high target expression (bottom). The curves show the population's expression distributions before selection, with the numbers above each curve indicating the selection round. The colored bars at the top indicate the FACS gates that were used to select cells from the populations at each corresponding round. (**C**) Examples of fluorescence distributions for individual clones obtained after five rounds of evolution. Microscopy pictures of two individual clonal promoter populations are shown as insets. (**D**) For each native *E. coli* promoter (blue) and synthetic promoter (red), the mean (x-axis) and variance (y-axis) of log-fluorescence intensities across cells were measured using flow cytometry. Fluorescence values are expressed in units of number of GFP molecules. The green curve shows the theoretically predicted minimal variance as a function of mean expression (Appendix 1). The insets show the log-fluorescence distributions for
*Figure 1. continued on next page*

*Figure 1. Continued*

two example promoters (corresponding to the larger dark blue and light blue dots). (**E**) Cumulative distributions of excess noise levels of native (blue) and synthetic (red) promoters.

The following figure supplements are available for figure 1:

**Figure supplement 1**. Genetic diversity of 378 sequenced promoters, which were extracted from randomly selected clones from the populations that were obtained after three and five rounds of selection.

**Figure supplement 2**. Mean log-fluorescence intensities as measured by FACS (horizontal axis) against estimated log GFP molecules per cell (vertical axis) as estimated from quantitative Westerns (see Appendix 1) for eight selected promoters.

**Figure supplement 3**. Relationship between log-protein levels as measured by GFP intensity in FACS (vertical axis) and log-mRNA levels (horizontal axis).

**Figure supplement 4**. Comparison of three biological replicate FACS measurements of means and excess noise of log-fluorescence for evolved *E. coli* promoters.

**Figure supplement 5**. Relative noise levels (variance of the log-expression distribution) of five pairs of native promoters that have very similar mean expression levels.

**Figure supplement 6**. Mean log-fluorescence (horizontal axis) and excess noise levels (vertical axis), that is, the difference between variance of log-fluorescence levels and the minimal variance at the corresponding mean, for all native (blue dots) and synthetic (red dots) promoters.

**Figure supplement 7**. Cumulative distributions of excess noise levels for the native (blue) and synthetic promoters (red).

We found, surprisingly, that most of the synthetic promoters exhibited noise levels close to the minimal level exhibited by the native promoters (*Figure 1D*). Additionally, a substantial fraction of native promoters exhibited excess noise levels significantly greater than the synthetic promoters (*Figure 1E* and *Figure 1—figure supplements 6, 7*). For example, only 26.1% of the synthetic promoters exhibited excess noise above 0.05, compared to 41.6% of the native *E. coli* promoters ($p < 7.7 \times 10^{-10}$, hypergeometric test). Given that the synthetic promoters were evolved from random sequence fragments and had not been selected on their noise properties (Appendix 2), we concluded that functional *E. coli* promoters should exhibit low excess noise levels by default. Importantly, this implies that the native promoters with elevated excess noise must have experienced selective pressures that caused them to increase their noise.

To understand how selection might have acted to increase noise, we first investigated whether excess noise was associated with other characteristics of the promoters. Previous studies in *Saccharomyces cerevisiae* have shown that promoters with high noise tend to also show high expression plasticity, that is, large changes in mean expression level across environments (*Newman et al., 2006*). Although we did not clearly observe this association in data from our previous study (*Silander et al., 2012*), a recent re-analysis of this data did uncover a significant association between expression plasticity and noise (*Singh, 2013*), which we confirmed using our present data (*Figure 2A*). In addition, we found that there is an equally strong relationship between excess noise and the number of regulators known to target the promoter (*Salgado et al., 2013*) (*Figure 2B*). In particular, whereas the excess noise levels of promoters without known regulatory inputs are very similar to those of our synthetic promoters, promoters with one or more regulatory inputs have clearly elevated noise levels (*Figure 2C*). The general association between elevated noise and gene regulation has recently been observed in eukaryotes as well (*Sharon et al., 2014*), and mutations that lower gene expression noise typically target TF binding sites (*Hornung et al., 2012*).

Our results imply that native promoters with high noise must have experienced selection pressures that caused their noise levels to increase, and that there is a general association between high noise and gene regulation. We next aimed to develop a theoretical understanding of these two observations. Perhaps the simplest interpretation of the observation that natural selection must have increased the noise levels of some promoters is that these promoters were directly selected for increased noise. Several theoretical treatments have shown that phenotypic variability may be

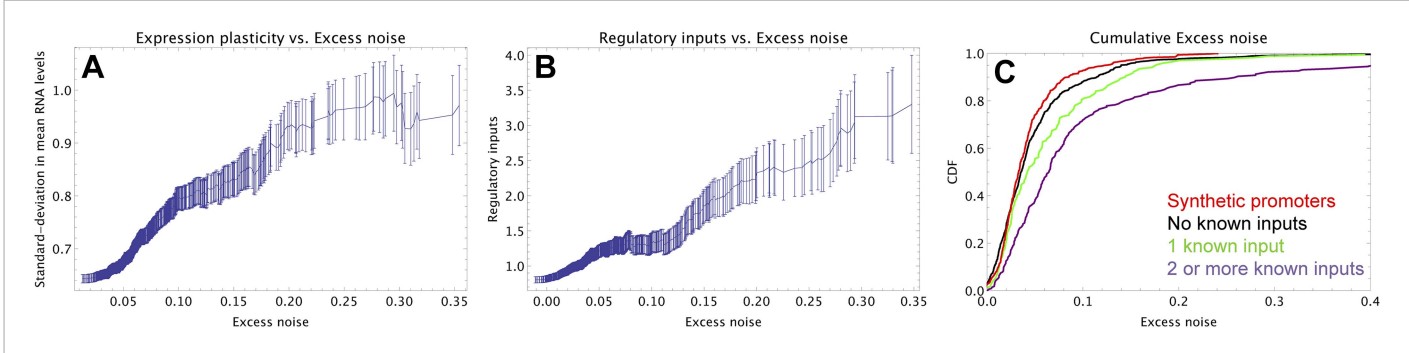

**Figure 2**. Promoters with elevated noise exhibit high expression plasticity and large numbers of regulatory inputs. (**A**) Native promoters were sorted by their excess noise $x$ and, as a function of a cut-off on $x$ (horizontal axis), we calculated the mean and standard error (vertical axis) of the variation in mRNA levels across different experimental conditions (data from http://genexpdb.ou.edu/) of all promoters with excess noise larger than $x$. (**B**) Promoters were sorted by excess noise $x$ as in panel **A**, and mean and standard error of the number of known regulatory inputs (vertical axis, data from RegulonDB [*Salgado et al., 2013*]) for promoters with excess noise larger than $x$ is shown. (**C**) Cumulative distributions of excess noise levels of synthetic promoters (red) and native promoters without known regulatory inputs (black), with one known regulatory input (green), and with two or more known regulatory inputs (purple).

selectively beneficial when environments change in ways that cannot be accurately sensed or are too rapid for organisms to respond (*Bull, 1987*; *Haccou and Iwasa, 1995*; *Kussell and Leibler, 2005*), with the phenotypic variability acting as a 'bet hedging' strategy. Thus, it is conceivable that selection has directly selected for increased noise as a bet hedging strategy for a subset of promoters, and more recent theoretical work shows that increasing gene expression noise may indeed increase population growth rates in some scenarios (*Tanase-Nicola and ten Wolde, 2008*). However, this interpretation does not explain the association between noise and regulation. On the contrary, one would naively expect that bet hedging strategies function as an alternative to gene regulation, that is, when implementing sensing and regulation would be either too difficult or too costly to evolve (*Kussell and Leibler, 2005*).

Regarding the general association between gene regulation and expression noise, using an analogy with the fluctuation-dissipation theorem from physics, it has been suggested that expression noise may be an unwanted but unavoidable side-effect of regulation (*Lehner and Kaneko, 2011*). Indeed, any regulator will have some noise in its expression or activity, and this noise will be propagated to its target genes. Consequently, this 'noise-propagation' effect will cause an increase in expression noise of the targets (*Thattai and van Oudenaarden, 2001*). Although noise-propagation is a plausible explanation for the general association between noise and regulation, its effects are detrimental to the accuracy of expression regulation, and one might thus expect natural selection to have acted to minimize its effects, for example, by minimizing the expression fluctuations in regulators. It would thus appear difficult to reconcile our observation that high noise promoters must have experienced selection to increase their noise levels with the assumption that selection has acted to minimize noise-propagation. Instead, our observations would be better explained by a scenario in which noise-propagation is positively selected for.

To clarify these observations, we developed a general theoretical model for quantifying how selection acts on gene regulatory interactions. In particular, the model calculates the effect on fitness of evolving a new regulatory interaction between a given gene and a given regulator, as a function of properties of the regulator, and the way selection acts on the gene's expression levels. As explained in 'Materials and methods' and Appendix 3, we derive that, under relatively mild assumptions, the fitness effects of a new regulatory interaction can be calculated analytically, and depend on only a few effective parameters. To explain this general model, we illustrate it using a simple scenario (*Figure 3*).

We focus on a single gene and assume that the gene starts out unregulated, with an expression distribution characterized by a certain mean $\mu$ and variance $\sigma^2$ (*Figure 3A*, blue curve). In its natural habitat, the population experiences a number of different environments $e$ that may require the gene to express at different levels and we assume that the fitnes of an individual cell, that is, its growth or survival rate, is a function of its gene expression state. Indeed, recent work has confirmed that

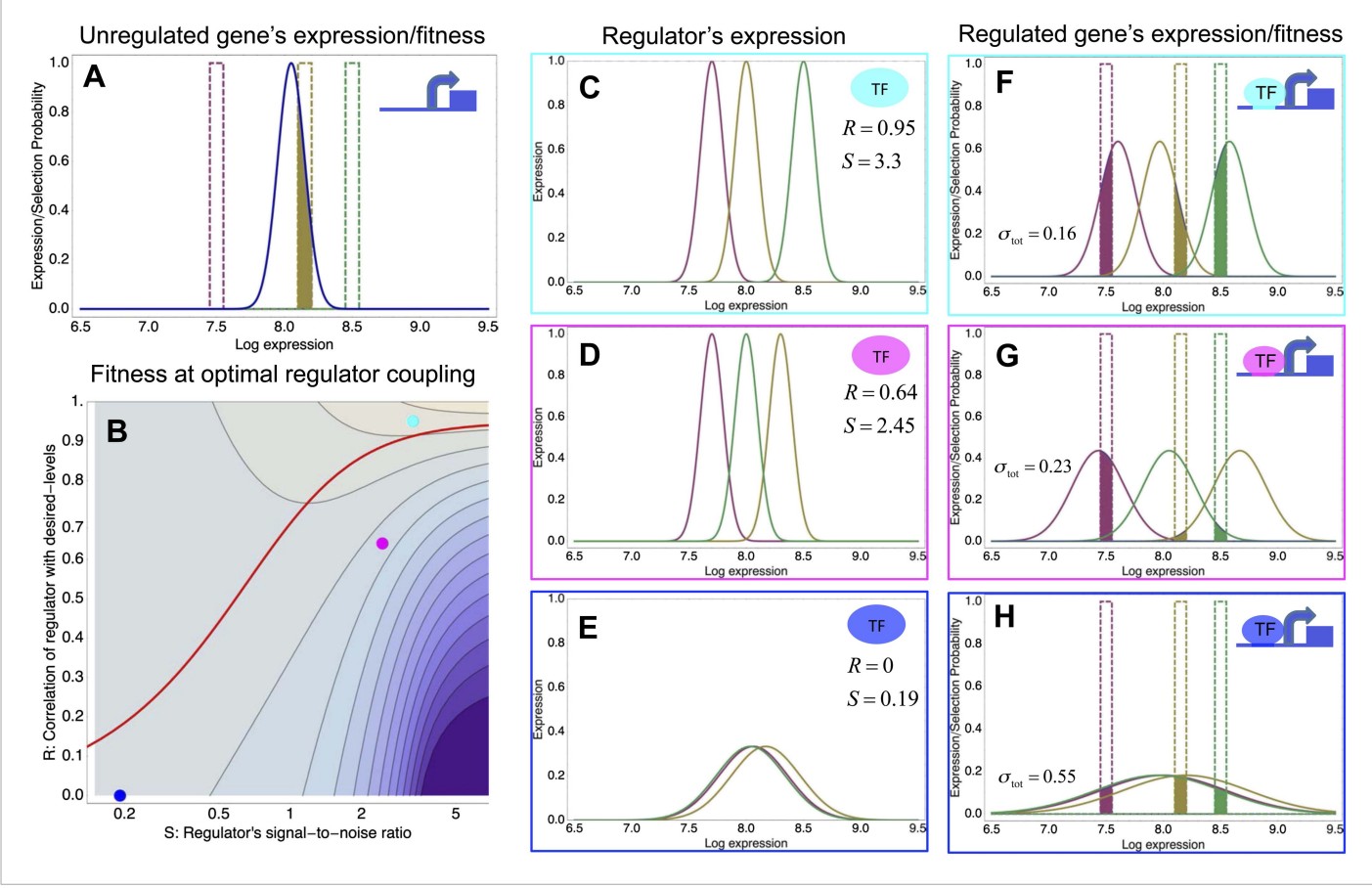

**Figure 3**. A model of the evolution of gene expression regulation in a variable environment. (**A**) Expression distribution of an unregulated promoter (blue curve) and selected expression ranges in three different environments, that is, the red, gold, and green dashed curves show fitness as a function of expression level in these environments. Although our model applies more generally, for simplicity we here visualize selection as truncation selection (i.e., a rectangular fitness function). The fitness of the promoter in the gold environment is proportional to the shaded area. (**B**) Contour plot of the log-fitness change resulting from optimally coupling the promoter to a transcription factor (TF) with signal-to-noise ratio $S$ and correlation $R$. Contours run from 7.5 at the top right to 0.5 at the bottom right. The three colored dots correspond to the TFs illustrated in panels **C–H**. The red curve shows optimal $S$ as a function of $R$. (**C–E**) Each panel shows the expression distributions of an example TF across the three environments (red, gold, and green curves). The corresponding values of correlation $R$ and signal-to-noise $S$ are indicated in each panel. (**F–H**) Each panel shows the expression distributions across the three environments for a promoter that is optimally coupled to the TF indicated in the inset. The shaded areas correspond to the fitness in each environment. The total noise levels of the regulated promoters are also indicated in each panel. The unregulated promoter has total noise $\sigma_{tot} = 0.1$.

The following figure supplements are available for figure 3:

**Figure supplement 1**. Phase diagram of the total noise $\sigma_{tot}$ of a promoter with expression mismatch $Y$ (horizontal axis) that is coupled (at optimal coupling strength) to a regulator whose regulatory activities have correlation $R$ with the desired expression levels (vertical axis) and whose signal-to-noise ratio $S$ has also been optimized.

**Figure supplement 2**. Inferred noise-propagation strengths of individual *E. coli* transcription factors (TFs).

expression fluctuations in single cells can affect their instantaneous growth rates (*Kiviet et al., 2014*). In the simple scenario of *Figure 3*, we assume there is an optimal level $\mu_e$ in each environment, and that cells with expression levels within a certain range $\tau$ around this optimum are selected. As an example, *Figure 3A* assumes there are three environments (red, gold, and green), with the green environment requiring up-regulation of the expression and the red environment requiring down-regulation of the expression. The fitness in each environment corresponds to the fraction of cells with expression levels within the selected range, that is, the unregulated promoter has reasonably high

fitness in the gold environment but very low fitness in the green and red environments. Since the overall fitness is the product of the fitness in each environment, a poor overlap between the expression distribution and selected range in any one environment leads to low overall fitness. In our model, the mismatch between the actual and desired expression levels is quantified by the 'expression mismatch' $Y$, where $Y^2 = \text{var}(\mu_e)/(\sigma^2 + \tau^2)$ is the variance in the desired expression levels $\mu_e$ across environments relative to the sum of the variances of the fitness function $\tau^2$ and the expression distribution $\sigma^2$ ($Y \approx 4$ for the example in *Figure 3A*).

We now consider evolving a regulatory interaction between the promoter and a given regulator. We assume that, in each environment $e$, the regulator's expression, or more generally its activity, will have some average level $r_e$. Coupling the promoter to the regulator will have two effects. First, the mean expression of the gene in each environment will become correlated with the mean activity of the regulator. Our model assumes a linear interaction, that is, in each environment $e$ the gene's mean expression becomes $\mu(e) = \mu + cr_e$, where $c$ is the coupling constant between regulator and promoter. This is the typical way in which we think about gene regulation and we will call this effect on the gene's mean expression the 'condition-response' effect. Second, in each environment $e$ the regulator's activity will also have some variance $\sigma_r^2$ and this noise will be propagated to the target gene. Because of this noise-propagation effect, the target's noise level will increase by $c^2\sigma_r^2$, becoming $\sigma^2 + c^2\sigma_r^2$. We define the renormalized coupling constant $X$ as the noise increase relative to the sum of the original noise levels and the variance of the fitness function, that is, $X^2 = c^2\sigma_r^2/(\sigma^2 + \tau^2)$.

Our analysis shows that, besides the expression mismatch $Y$ and coupling constant $X$, the fitness increase that results from coupling to a given regulator depends only on two effective parameters characterizing the regulator: First, the Pearson correlation coefficient $R$ between the desired expression levels $\mu_e$ and the regulator's average activities $r_e$ and, second, the signal-to-noise ratio of the regulator $S$, with $S^2 = \text{var}(r_e)/\sigma_r^2$ defined as the ratio of the variance in the mean activities of the regulator across the environments and the noise level of the regulator in each environment. In terms of these parameters, the increase in log-fitness resulting from evolving a regulatory interaction becomes

$$d\log[f(X, Y, R, S)] = \frac{1}{2}\frac{(X^2 + R^2)Y^2 - (SX - RY)^2}{1 + X^2} - \frac{1}{2}\log[1 + X^2]. \qquad (1)$$

Notably, this equation applies independent of how the desired levels $\mu_e$ and regulator levels $r_e$ vary across the environments and only depends on the assumption that the fitness function and expression distributions can be approximated by Gaussians.

To illustrate the predictions of this theory, the contour plot of *Figure 3B* shows the log-fitness changes that can be obtained by optimally coupling the promoter to a regulator with a given correlation $R$ and signal-to-noise $S$. We chose the range in $S$ values such that $d\log[f]$ is positive over the parameter region shown, that is, $d\log[f] \approx 0$ in the lower right corner of *Figure 3B*. As intuitively expected, the highest fitness is obtained when coupling to an accurate regulator with high signal-to-noise $S$, whose activities correlate precisely with the desired expression levels (cyan dot in *Figure 3B*). An example of such a TF is shown in *Figure 3C*. The resulting expression distributions of the promoter coupled to this TF accurately track the desired levels, with only moderately increased noise in the promoter's expression (*Figure 3F*). In this parameter regime, the improvement in fitness is entirely due to the condition-response effect, and the increased noise of the target can indeed be considered a detrimental side-effect of the regulation.

However, regulators that track the desired expression levels of the promoter with such high accuracy, that is, $R = 0.95$, may often not be available. Interestingly, coupling to a noisy regulator whose activity is entirely uncorrelated with the desired expression levels (blue dot in *Figure 3B* and *Figure 3E,H*) also substantially increases fitness. In this regime, the increased fitness results exclusively from the noise-propagation mechanism, and by coupling to the regulator the promoter effectively implements a bet hedging strategy.

Surprisingly, coupling to the uncorrelated noisy regulator (blue dot in *Figure 3B* and *Figure 3E,H*) outperforms coupling to a moderately correlated regulator (magenta dot in *Figure 3B* and *Figure 3D, G*). To understand how coupling to a regulator with moderate correlation $R = 0.64$ can be outperformed by coupling to a regulator with no correlation $R = 0$, we calculated the optimal signal-to-noise $S$ as a function of its correlation $R$ (red curve in *Figure 3B*). This shows that the magenta regulator has too large an $S$ for its correlation, that is, increasing the noise of this TF would result in an

increase of the promoter's noise, and this would in turn lead to an increase in fitness in the green and gold conditions (see *Figure 3G*). This illustrates that regulators may generally be under selection to become noisy themselves. To the left of the red curve in *Figure 3B*, noise-propagation is too large and the increased noise of the targets can be considered a detrimental side-effect of regulation. In contrast, to the right of the red curve, noise-propagation is too small, that is, increasing the noise of the regulator would improve fitness.

Most interestingly, the red curve corresponds to a continuum of regulatory strategies in which the condition-response and noise-propagation effects are optimally acting in concert, going from being dominated by noise-propagation in the lower left, to being dominated by the condition-response in the top right. Importantly, this clarifies how accurate regulation can evolve smoothly from a state without regulation. Highly accurate regulation with high $R$ and $S$ can be reached by starting from coupling to a noisy regulator with low $R$ and $S$, whose benefits come entirely from the noise-propagation, and then increasing both $R$ and $S$ in incremental steps along this continuum of regulatory strategies.

We now discuss how this theoretical model helps us interpret our experimental observations. First, our model predicts that the selective pressure for a promoter to evolve regulatory interactions is determined by the expression mismatch $Y$. When $Y < 1$, even a constitutively expressed promoter has a good overlap with the fitness function across all conditions, and there will be no selective pressure to evolve regulation. Our synthetic promoters, which were selected for expressing at a constant level, correspond to this situation, and our results show that such promoters have low noise by default. Thus, we interpret the observation that native promoters without known regulatory inputs have noise levels similar to those of our synthetic promoters (*Figure 2B*) as indicating that constitutive promoters have low noise by default.

In the interpretation of our model, the promoters with elevated noise were those in which selection required their expression levels to vary significantly across environments, that is, for which $Y \gg 1$. How much expression noise a given promoter is likely to evolve depends on its value of its expression mismatch $Y$, and the values of the correlations $R$ and signal-to-noise $S$ of the regulators that are available in the genome. Since the precise environmental conditions that *E. coli* experiences in the wild, and how these determine optimal expression levels of its genes, are largely unknown, it is not possible to make quantitative predictions of the expected noise levels of specific promoters using our model. However, our model can be used to understand the general qualitative trends observed in *Figure 2*.

First, our model explains why there is a general correlation between expression noise and expression plasticity. Since regulators affect the expression of their targets in a linear manner, coupling a promoter to a combination of different regulators is equivalent to coupling the promoter to a single 'effective regulator' whose expression distribution is a linear combination of the expression distributions of the individual regulators. Assuming that coupling to such a linear combination of regulators can attain a correlation $R$ with the desired expression levels of the gene, our model predicts that noise-propagation will be selected whenever $Y^2 > (1 - R^2)^{-1}$; that is, whenever the expression mismatch $Y$ is large enough, noise-propagation will be beneficial. If we additionally assume that selection has tuned the signal-to-noise of the regulators to optimize the amount of noise-propagation, then the final noise level of the promoter is predicted to equal $\sigma_{\text{tot}}^2 = (1 - R^2)\text{var}(\mu_e) - \tau^2$ (*Figure 3—figure supplement 1* and Appendix 3). This expression can be interpreted as saying that, of the original mismatch $Y^2$, a fraction $R^2 Y^2$ is accounted for by the condition-response, whereas the remaining fraction $(1 - R^2)Y^2$ is accounted for by the noise-propagation. This implies that both the expression plasticity, which is given by the variance in the promoter's mean across conditions (i.e., $R^2 Y^2$), and the noise level (i.e., $(1 - R^2)Y^2$) are proportional to the original expression mismatch $Y^2$. Our model thus predicts that the expression plasticity and noise level should be correlated.

Our model also predicts a general positive correlation between expression noise and the number of regulatory inputs. Starting from a high expression mismatch $Y$, each new regulatory interaction will reduce the mismatch from $Y$ to $Y' < Y$ by a combination of the condition-response effect reducing the average deviations from the desired levels, and the noise-propagation increasing the overlap by virtue of increasing the expression noise. Whenever $Y'$ is still larger than 1, the promoter will be under selective pressure to evolve further regulatory interactions. In this way, the higher the initial mismatch $Y$, the larger the expected number of regulatory interactions that will be necessary to reduce the mismatch below 1; that is, our model generally predicts that the number of regulatory interactions, the expression plasticity, and the final noise all correlate with the original mismatch $Y$.

Finally, since in our model elevated noise levels are due to noise-propagation per definition, it trivially predicts that, the larger the number of regulatory inputs, the larger the final noise levels tends to be. More specifically, our model predicts that, in a given condition, the noise level of a promoter is determined by the noise levels of the TFs that regulate it. To test this prediction, we used a very simple linear model that assumes that the excess noise level of a gene is equal to the sum of the noise levels of the TFs that regulate it ('Materials and methods'). Although this simple model is very crude, that is, assuming noise-propagation to be of equal size for all targets of a given TF, and assuming that fluctuations in TF activities are all independent, it was nevertheless able to explain a substantial fraction of the variance in excess noise levels across promoters (17%). The top five TFs most significantly associated with elevated noise levels of their targets were CRP, H-NS, ArcA, ilvY, and GadX (*Figure 3—figure supplement 2*). Of these, GadX and H-NS were also identified, using a simpler method, in the data of our previous study (*Silander et al., 2012*). The appearance of H-NS is interesting since it is a histone-like nucleoid-associated protein that acts as a silencer (*Dorman, 2004*), that is, somewhat analogous to the role of nucleosomes in eukaryotes, and in eukaryotes nucleosome positioning at promoters has been shown to be a major determinant of transcriptional noise (*Blake et al., 2006*; *Tirosh and Barkai, 2008*; *Cairns, 2009*). The TFs ArcA and GadX are involved with responses to low oxygen and acid stress, respectively, and it is plausible that these TFs may be partially activated in the conditions in which our experiments are performed. Our cells are grown in M9 minimal media with glucose in micro-titer plates, and measurements are taken late in the exponential phase. It is well-known that in micro-titer plates oxygen limitation can become a major stress late in the exponential phase, and this may result in the activation of fermentation reactions which in turn cause acid stress. The appearance of CRP is consistent with our observation in *Silander et al. (2012)* that promoters of genes involved in carbon metabolism are over-represented among high noise promoters. In summary, modeling of excess noise levels in terms of known regulatory interactions shows that, in accordance with our model, a substantial amount of the variation in noise levels can be explained by noise-propagation from noisy regulators, and the regulators we identify as most significantly propagating noise are consistent with existing biological knowledge regarding our growth conditions.

## Discussion

Because genotype-phenotype relationships for complex phenotypic traits are poorly understood, it is often difficult to assess how observable variation in a particular trait has been affected by natural selection. Here we have shown that, by comparing naturally observed variation in a particular trait with variation observed in synthetic systems that were evolved under well-controlled selective conditions, definite inferences can be made about the selection pressures that have acted on the natural systems. In particular, by evolving synthetic *E. coli* promoters de novo using a procedure in which promoters are strongly selected on their mean expression and not on their expression noise, we have shown that native promoters must have experienced selective pressures that increased their noise levels, and that promoters with elevated noise are highly regulated by TFs.

To account for this, we have developed a theoretical model that provides a simple mechanistic framework for understanding how selection acts on regulatory interactions. The key ingredient of the model is that it recognizes that a regulatory interaction affects the target's expression in two separate ways: the condition-response effect through which the mean expression of the target becomes a function of the mean activity of the regulator, and the noise-propagation effect through which the noise of the target is increased in proportion to the noise of the regulator. Our model elucidates that not only the condition-response effect but also the noise-propagation effect is often a *functional* consequence of the regulatory interaction; that is, instead of being just an unavoidable side-effect of regulation, noise-propagation is often beneficial and can be considered to act as a rudimentary form of regulation. Our framework vastly expands the evolutionary conditions under which novel regulatory interactions can evolve. Instead of assuming that regulators and their targets must evolve in a tightly coordinated fashion, noise-propagation alone may provide a sufficient benefit for a new regulatory interaction to evolve. This regulation can then be smoothly mutated along a continuum in which noise-propagation and condition-response are acting in concert, slowly lowering noise, and increasing the accuracy of the condition-response, eventually leading to highly accurate regulation. In this way our model provides a plausible scenario for how accurate regulatory interactions can evolve de novo from a state without regulation. Finally, our model shows that unless regulation is very precise, regulatory

interactions that act to increase noise are beneficial. Thus, elevated levels of expression noise can be expected whenever the accuracy of regulation is limited.

## Materials and methods

### *Ab initio* promoter library construction from random sequences

We obtained chemically synthesized nucleotide sequences of random nucleotides 200 bp in length (Purimex, Germany). Each sequence had defined 5′ and 3′ ends to allow PCR amplification. Within these constant regions, restriction sites for BamHI and XhoI were present. The intervening sequence was made up of 157 bp of random nucleotides (5′-CCTTTCGTCTTCACCTCGAG-(N157)-GGGATCCTCTGGATGTAAGAAGG-3′). However, as coupling of base pairs during oligonucleotide synthesis is not always successful and strand breaks can frequently occur in long oligonucleotides, many oligonucleotides were shorter than 200 bp in length. We used PCR to generate double-stranded DNA from the single-stranded oligonucleotides using forward and reverse primers matching the defined 5′ and 3′ ends. We gel-purified the double-stranded PCR product and double-digested it using BamHI and XhoI. After column purification, sequences were ligated into a version of the low-copy plasmid pUA66, which contains a gfpmut2 open reading frame downstream of a strong ribosomal binding site (*Zaslaver et al., 2006*). The vector was modified to remove a weak $\sigma$70 binding site present 24 bp upstream of the GFP open reading frame (two point mutations, A → G and T → G, were introduced, changing the putative $\sigma$70 binding site from TAGATT to TGGATG, with the consensus $\sigma$70 binding site being TATAAT). The ligation was performed using T4 DNA ligase (NEB) at 16°C for 24 hr. The ligation product was then column purified and electroporated into *E. coli* DH10B cells. This protocol resulted in extremely high transformation yields (approximately $10^6$ individual clones per transformation).

### Selection on expression level using flow cytometry

Cultures of transformed cells were regenerated for 1 hr in 1 ml SOC medium (Super Optimal Broth supplemented with 20 mM glucose) and afterwards 1 ml SOC containing 50 μg/ml kanamycin was added for overnight growth, ensuring that only cells containing the plasmid could grow. These cultures were then diluted 500-fold (approximately $5 \times 10^6$ cells in total) into M9 minimal media supplemented with 0.2% glucose and grown for 2.5 hr with shaking at 200 rpm. The distribution of GFP fluorescence levels was measured for each culture using FACS in a FACSAria IIIu (BD Biosciences), with excitation at 488 nm and a 513/17 nm bandpass filter used for emission.

We used this distribution of fluorescence values to designate a selection gate. The position of the gate was determined by measuring the mean fluorescence of two reference promoters (*Zaslaver et al., 2006*): *gyrB* which exhibits a mean expression level that is at the 50th percentile all *E. coli* promoters; and *rpmB*, which exhibits a mean expression level that is at the 97.5th percentile of all *E. coli* promoters (*Silander et al., 2012*). For each of these reference genes, the mean fluorescence level was measured and a selection gate was constructed, centered on this mean expression level, such that 5% of all clones in the population fell within the gate. For each round of selection, we sorted 200,000 cells contained within this gate. Sorted cells were then transferred to 4 ml Luria Broth (LB) media (containing 50 μg/ml kanamycin) and grown overnight. These cultures were stored supplemented with 7.5% glycerol at −80°C for subsequent analysis.

For each expression level (i.e., reference gene) we evolved three replicate populations. We refer to these as the medium expressers (those promoters selected based on the *gyrB* reference gate) and high expressers (those promoters selected based on the *rpmB* reference gate).

### PCR mutagenesis

Following FACS-based selection on fluorescence, we introduced novel genetic variation into the populations using PCR mutagenesis. We first re-grew the cells overnight and used this culture to prepare plasmid DNA. We amplified the promoter sequences from these plasmids using the GeneMorph II Random Mutagenesis Kit (Stratagene) with the primers referred to previously that matched the defined regions of the promoters. We used 0.01 ng of DNA as starting material and 35 cycles for amplification. This resulted in a mutation rate of around 0.01 per bp (such that we expect that, in 200 bp, 95% of the promoters will contain between zero and four mutations). These PCR products were then digested with XhoI and BamHI, ligated back into the vector, and again

transformed into DH10B cells. After an initial round of selection on the initial library, this entire process (PCR mutagenesis, transformation, and selection) was repeated four times in total. At this point, the plasmid libraries of synthetic promoters were isolated and transformed into *E. coli* K12 MG1655 for comparison with a library of native *E. coli* promoters (see below).

## Quantification of fluorescence

To quantify fluorescence on a single-cell level, we used flow cytometry with a FACSCanto II (BD Biosciences), with excitation at 488 nm and a 513/17 nm band-pass filter used for emission. We collected data for at least 50,000 events. We then gated this data as outlined in *Silander et al. (2012)*, identifying approximately 5000 cells most similar in forward scatter (FSC) and side scatter (SSC). We then calculated the mean and variance in log-fluorescence using these cells, using a Bayesian procedure that accounts for outliers (Appendix 1). We randomly selected 479 promoters from the evolved set (72 medium expressers and 72 high expressers after three rounds of selection; 168 medium expressers and 167 high expressers after five rounds of selection) and quantified mean and variance in fluorescence. We used the same measurement procedures to calculate mean and variance for all promoters contained in a library of *E. coli* promoters also placed upstream of the gfpmut2 open reading frame on the pUA66 plasmid (*Zaslaver et al., 2006*). We refer to the promoters from this library as native *E. coli* promoters. For 288 promoters, we quantified fluorescence in three independent cultures and found that both mean and variance in expression were reproducible across replicate biological experiments (*Figure 1—figure supplement 4*). Additionally, we sequenced 378 sequences from our set of 479 promoter sequences, which showed that even after five rounds of selection, the promoters were quite diverse (*Figure 1—figure supplement 1*). To confirm the sensitivity and accuracy of the FACS measurements, we selected 10 promoters and used fluorescence microscopy to measure their mean and variance in fluorescence. The cells were grown in the same conditions described above, placed on 1% agarose pad, and images were obtained using a CoolSNAP HQ CCD camera (Photometrics) connected to a DeltaVision Core microscope (Applied Precision) with a UPlanSApo 100×/1.40 oil objective (Olympus). Image-processing was done in soft-WoRx v3.3.6 (Applied Precision) and fluorescence values were extracted based on DIC image-mediated cell detection in MicrobeTracker Suite (*Sliusarenko et al., 2011*). For each cell, we calculated fluorescence per cell volume by summing all pixel values and dividing by the volume of the cell as estimated by MicrobeTracker. Cells undergo substantial phenotypic changes when they are put on agar, including changes in the distribution of cell sizes. Consequently, it is problematic to compare absolute variance measurements directly between FACS and microscope. We therefore compared the relative noise levels of different promoters. The 10 selected native promoters consist of five pairs with almost identical mean expression values (as measured by FACS) but with noise levels that vary by different amounts. For each of the five pairs we calculated the ratio of the noise levels of the higher and the lower noise promoter as measured by both FACS and the microscope. As shown in *Figure 1—figure supplement 5*, with the exception of one pair of promoters that showed almost equal noise levels in FACS but a 50% difference in noise in the microscope, all other pairs showed good correlation of the relative noise levels in FACS and in the microscope, confirming that relative noise levels are similar in FACS and microscope measurements.

## Quantitative Western analysis

To determine the correspondence between fluorescence intensities and absolute GFP numbers per cell, eight individual promoter clones were grown in three biological replicates using the same media conditions as in the experimental evolution. The cells were then re-suspended in SDS sample buffer, heated for 5 min at 95°C, and proteins were resolved by 12% SDS-PAGE. Quantification was done by loading a standard curve consisting of 10, 25, 50, 75, and 100 ng of GFP (#632373; Clonetech). Proteins were transferred to a Hybond ECL membrane (GE Healthcare, Life Sciences), which was then blocked in TNT (20 mM Tris pH 7.5, 150 mM NaCl, 0.05% Tween 20) with 1% BSA and 1% milk powder. Detection was performed with the ECL system after incubation with rabbit anti-GFP and polyclonal pig anti-rabbit. Western intensities for each sample were extracted using ImageJ (*Figure 1—figure supplement 2*). The number of cells loaded was estimated by calculating the relationship between OD600 and CFU counts. Details of the data analysis procedures are given in Appendix 1.

## Correlating protein and RNA levels per cell by quantitative PCR

Native and evolved single-promoter populations were grown in three biological replicates by diluting overnight LB cultures 500-fold into M9 media supplemented with glucose. These cultures were grown for 2.5 hr, stabilized with an equal volume of RNA Later (Sigma–Aldrich) and RNA was extracted using the Total RNA Purification 96-Well Kit (Norgen Biotek Corp) with on-column DNAse I digestion. Reverse transcription was done using random hexamers and qPCR with TaqMan probes and performed by Eurofins Medigenomix GmbH (Germany). Three technical replicates were performed. The efficiency of the primers and probes used were validated in a dilution series. Relative RNA levels per cell were obtained by normalizing to the reference gene *ihfB* using a Bayesian procedure for integrating data from the replicates and accounting for failed measurements (Appendix 1). The primers and probes used were: GFP forward primer: 5′-CCTGTCCTTTTACCAGACAA-3′; GFP reverse primer: 5′-GTGGTCTCTCTTTTCGTTGGGAT-3′; GFP probe: 5′-TACCTGTCCACACAATCTGCCCTTTCG-3′, ihfB forward primer: 5′-GTTTCGGCAGTTTCTCTTTG-3′, ihfB reverse primer: 5′-ATCGCCAGTCTTCGGATTA-3′, ihfB probe: 5′-ACTACCGCGCACCACGTACCGGA-3′).

## Minimal variance as a function of mean expression and excess noise

In a simple model of gene expression in which there are constant rates of transcription, translation, mRNA decay, and protein decay, the probability distribution for the number of proteins per cell is a negative binomial with variance proportional to the mean $\langle n \rangle$: $\mathrm{var}(n) = (b + 1)\langle n \rangle$, where the constant $b$ is the ratio between the mRNA translation rate and the mRNA decay rate, which is often referred to as 'burst size' (*Shahrezaei and Swain, 2008*). However, in general there are also cell-to-cell fluctuations in the transcription, translation, and decay rates, which are proportional to these rates themselves. These fluctuations lead to an additional term in the variance $\mathrm{var}(n)$ which is proportional to the square of the mean: $\mathrm{var}(n) = \beta \langle n \rangle + \sigma_{ab}^2 \langle n \rangle^2$, where $\beta$ is a renormalized burst size and $\sigma_{ab}^2$ is the relative variance of the product of transcription, translation, and decay rates across cells (Appendix 1).

The total fluorescence in a cell (measured in units equivalent to number of GFP proteins) $n_{\mathrm{meas}}$ can then generally be written as: $n_{\mathrm{meas}} = n_{\mathrm{bg}} + \langle n \rangle + \epsilon \sqrt{\mathrm{var}(n)}$, where $n_{\mathrm{bg}}$ is background fluorescence and $\epsilon$ is a fluctuating quantity with mean zero and variance one. Assuming that the fluctuations are small relative to the mean, we then find for the variance of the logarithm of $n_{\mathrm{meas}}$:

$$\mathrm{var}(\log[n_{\mathrm{meas}}]) = \sigma_{ab}^2 \left(1 - \frac{n_{\mathrm{bg}}}{\langle n_{\mathrm{meas}} \rangle}\right)^2 + \frac{\beta}{\langle n_{\mathrm{meas}} \rangle} \left(1 - \frac{n_{\mathrm{bg}}}{\langle n_{\mathrm{meas}} \rangle}\right).$$

We fit this functional form to the minimum variance $\mathrm{var}(\log[n_{\mathrm{meas}}])$ as a function of the mean, with $\sigma_{ab}^2 = 0.025$ and $\beta = 450$. We defined the excess variance as the difference between the measured variance and this fitted minimal variance. A more detailed derivation is given in Appendix 1.

## The FACS selection function

By comparing the distributions of the population's expression levels before and after rounds of selection (without intervening mutation of the promoters), we found that the probability that a cell with expression level $x$ is selected by the FACS is well-approximated by $f(x|\mu_\star, \tau) = \exp\left[-\frac{(x - \mu_\star)^2}{2\tau^2}\right]$, with $\mu_\star$ the desired expression level and $\tau$ the width of the selection window. For the last three rounds of selection for medium expression, the selection gates in the FACS were relatively constant, and we estimated $\tau \approx 0.03$ and $\mu_\star$ fluctuated slightly around an average value of $\mu_\star \approx 8.1$ for these selection rounds.

With this selection function, a promoter genotype that exhibits a distribution of expression values with mean $\mu$ and standard deviation $\sigma$ has a fitness (fraction of cells selected in the FACS) of

$$f(\mu, \sigma | \mu_\star, \tau) = \sqrt{\frac{\tau^2}{\tau^2 + \sigma^2}} \exp\left[-\frac{(\mu - \mu_\star)^2}{2(\tau^2 + \sigma^2)}\right]. \tag{2}$$

This estimated fitness function indicated that the fitness of promoter genotypes strongly depends on their mean $\mu$ and is almost independent of their excess noise (*Appendix 2—figure 3* and *Appendix 2—figure 4*). In addition, applying additional rounds of selection of varying strengths to the population of evolved promoters did not systematically alter their distribution of excess noise levels. Details of the analysis of the FACS selection are given in Appendix 2.

## Model for the evolution of gene regulation in a fluctuating environment

Although the model we present can be extended to include the evolution of gene regulation for multiple genes, for simplicity we focused on the evolution of a single gene and its promoter. We assume that the population experiences a sequence of different environments and that, in each environment, the fitness of each organism is a function of its gene expression level. We characterized the fitness function in each environment by two parameters: the desired level $\mu_e$ that maximizes the fitness and a parameter $\tau$ that quantifies how quickly fitness falls away from this optimum. For simplicity and analytical tractability, we assumed a Gaussian form: $f(x|\mu_e, \tau) = \exp\left[-\frac{(x-\mu_e)^2}{2\tau^2}\right]$. Similarly, although it is straightforward to allow the variance $\tau^2$ to vary across conditions, the results are more transparent when we assume $\tau^2$ is the same in all environments. Note that this fitness function has the same form as the FACS selection function. Consequently, the fitness $f(\mu, \sigma|\mu_e, \tau)$ of a promoter with mean $\mu$ and variance $\sigma^2$ is given by *Equation 2* as well, with $\mu_e$ replacing $\mu_\star$.

The total number of offspring that a promoter will leave behind after experiencing all environments is given by the product of its fitness in each of the environments. Equivalently, the log-fitness of a promoter is proportional to its average log-fitness across all environments. For an unregulated promoter with fixed mean $\mu$ and variance $\sigma^2$ in expression, we then find for the log-fitness:

$$\log\left[f(\mu, \sigma)\right] = -\frac{(\mu - \langle\mu_e\rangle)^2 + \text{var}(\mu_e)}{2(\tau^2 + \sigma^2)} + \frac{1}{2}\log\left[\frac{\tau^2}{\tau^2 + \sigma^2}\right],$$

where $\langle\mu_e\rangle$ is the average of the desired expression levels across environments and $\text{var}(\mu_e)$ is the variance in the desired expression levels across environments. If we do not consider gene regulation but simply optimize the promoter's mean expression and noise level, then we find optimal log-fitness occurs when $\mu = \langle\mu_e\rangle$ and $\sigma^2 = 0$ (when $\text{var}(\mu_e) < \tau^2$) or $\sigma^2 = \text{var}(\mu_e) - \tau^2$ otherwise. That is, when the desired expression level varies more than the width of the selection window, fitness is optimized by increasing noise so as to ensure the distribution overlaps the desired levels across all conditions. This result is equivalent to previous results on the evolution of phenotypic diversity in fluctuating environments (*Bull, 1987*).

To increase fitness, a promoter can evolve to become regulated by one of the regulators existing in the genome. Instead of having a constant mean expression $\mu$, the promoter's mean expression will then become a function of the environment $e$: $\mu(e) = \mu + cr_e$, where $r_e$ is the mean expression (or more generally regulatory activity) of the regulator in environment $e$, and $c$ is the coupling strength. Note that, for simplicity, we thus assume a linear coupling between the means of regulator and target. Since any gene will have some variability in its expression, we assumed that the actual expression/activity of the regulator in each environment $e$ is Gaussian distributed with a variance $\sigma_r^2$. As for the width of the fitness function $\tau$, it would again be straightforward to allow $\sigma_r^2$ to vary across conditions (as it likely does in reality). However, the results are analytically more transparent and bring out the main features of the model better if we assume the regulator's noise $\sigma_r^2$ is the same in all conditions.

When coupled to the regulator, the promoter's total expression variance will become $\sigma_{\text{tot}}^2 = \sigma^2 + c^2\sigma_r^2$ and the log-fitness of the promoter becomes:

$$\log\left[f(\mu, \sigma, c)\right] = -\frac{\langle(\mu + cr_e - \mu_e)^2\rangle}{2(\tau^2 + \sigma^2 + c^2\sigma_r^2)} + \frac{1}{2}\log\left[\frac{\tau^2}{\tau^2 + \sigma^2 + c^2\sigma_r^2}\right].$$

Assuming that the basal expression level $\mu$ is optimized to maximize log-fitness, that is, $\mu = \langle\mu_e\rangle - c\langle r_e\rangle$, this log-fitness can be rewritten as:

$$\log\left[f(X, Y, S, R)\right] = \text{cons.} - \frac{1}{2}\frac{Y^2(1 - R^2) + (SX - RY)^2}{1 + X^2} - \frac{1}{2}\log\left[1 + X^2\right].$$

where $X$ measures the coupling strength $\left(X^2 = \frac{c^2\sigma_r^2}{\tau^2 + \sigma^2}\right)$, $Y$ is the expression mismatch that measures how much the desired expression level varies across environments $\left(Y^2 = \frac{\text{var}(\mu_e)}{\tau^2 + \sigma^2}\right)$, $S$ is the signal-to-noise of the regulator $\left(S^2 = \frac{\text{var}(r_e)}{\sigma_r^2}\right)$, and $R$ is the Pearson correlation between the desired expression levels $\mu_e$ and the activity levels $r_e$ of the regulator. The change in log-fitness between the situation

before and after adding of the regulatory interaction is obtained by subtracting $\log[f(0, Y, S, R)]$ from $\log[f(X, Y, S, R)]$, yielding

$$d \log\left[f(X, Y, S, R)\right] = \frac{1}{2}\frac{Y^2\left(R^2 + X^2\right) - (SX - RY)^2}{1 + X^2} - \frac{1}{2}\log\left[1 + X^2\right].$$

Note that this basic argument can be iterated. After the promoter has been coupled to a regulator, the residual deviations between the desired and actual expression levels are given by $\tilde{\mu}_e = \mu_e - cr_e$ and the new noise level of the promoter is given by $\tilde{\sigma}^2 = \sigma^2 + c^2\sigma_r^2$. If we define a new expression mismatch $\tilde{Y}^2 = \mathrm{var}(\tilde{\mu}_e)/(\tilde{\sigma}^2 + \tau^2)$, then we can calculate the log-fitness changes associated with adding another regulatory interaction using exactly the same expressions as above, replacing $Y$ by $\tilde{Y}$.

In addition, because the coupling between the activity of the regulator and the expression of the promoter is linear, coupling the promoter to an arbitrary linear combination of different regulators can be modeled as coupling the promoter to a single 'effective' regulator; that is, if a promoter is coupling to different regulators $r^i$ with coupling constants $c_i$, then in environment $e$ we have $\mu(e) = \mu + \sum_i c_i r_e^i$, which is equivalent to coupling with constant $c$ to a regulator with mean $r_e = \sum_i c_i r_e^i/c$. If $\sigma_i^2$ is the noise level of regulator $i$ and $R_{ij}$ is the Pearson correlation in the fluctuations of regulators $i$ and $j$, then this composite regulator has a total variance $\sigma_r^2 = \sum_i c_i^2\sigma_i^2 + \sum_{i \neq j} R_{ij}\sigma_i\sigma_j$.

As can be easily seen from *Equation 1* in the main text, if the best linear combination of regulators provides a correlation $R$ with the promoter's desired levels, the optimal value $S_*$ of the signal-to-noise of this composite regulator is given by $S_* = RY/X$. Substituting this back into *Equation 1*, we find for the optimal coupling strength $X_*^2 = \max[0, (1 - R^2)Y^2 - 1]$. This function is plotted in *Figure 3—figure supplement 1*, together with the values of $S_*$ as a function of $Y$ and $R$. Note that $(1 - R^2)Y^2$ is the part of the expression mismatch that is not accounted for by the condition-response effect of the regulators. Whenever this remaining expression mismatch is less than 1, noise-propagation is a detrimental side-effect of regulation and regulators will be selected to be as accurate as possible. However, when $(1 - R^2)Y^2 > 1$, noise-propagation will be selected for, and the increase in the total noise is equal to the amount of expression mismatch not accounted for by the condition-response.

Additional details on the derivation of our model and analysis of the behavior of the fitness function as a function of its parameters are given in Appendix 3.

## Analysis of excess noise against gene expression variation and regulatory inputs

We re-annotated the promoter fragments of *Zaslaver et al., 2006* by mapping the published primer pairs to the *E. coli* K12 MG1655 genome. Of the 1816 promoter fragments, 1718 could be unambiguously associated with a gene that was immediately downstream, and the 1718 promoter fragments were associated with 1137 different downstream genes (for some genes there were multiple or repeated upstream promoter fragments). We used the operon annotations of RegulonDB (*Salgado et al., 2013*) to extract, for each promoter, the set of additional downstream genes that are part of the same operon as the first downstream gene. We obtained known regulatory interactions between TFs and genes from RegulonDB and counted, for each *E. coli* gene, the number of TFs known to regulate the gene. We defined the number of regulatory inputs of a promoter to equal the average of the number of inputs for all genes in the operon downstream of the promoter. We sorted promoters by their excess noise and, as a function of a cut-off on excess noise level, calculated the mean and standard error of the number of regulatory inputs for all promoters with excess noise level above the cut-off. We obtained genome-wide gene expression measurements from the Gene Expression Database (http://genexpdb.ou.edu/). For each *E. coli* gene, we obtained 240 log fold-change values $x$ corresponding to the logarithm of the expression ratio of the gene in a perturbed and a reference condition. We defined the variance in expression of a gene as the average of $x^2$ across the 240 experiments. We again sorted promoters by their excess noise and, as a function of a cut-off on excess noise level, calculated the mean and standard error of gene expression variances for all promoters with excess noise above the cut-off.

## Fitting excess noise levels in terms of regulatory interactions

Using the RegulonDB database (*Salgado et al., 2013*), we constructed a binary matrix **R** of regulatory interactions, where the components $R_{pr} = 1$ when regulator $r$ is known to target promoter $p$, and $R_{pr} = 0$

otherwise. Following previous work from our group in which we modeled gene expression patterns in mammals in terms of regulatory sites (*FANTOM Consortium et al., 2009*; *Balwierz et al., 2014*), we use a simple linear model to relate the excess noise $E_p$ of each promoter $p$ to the (unknown) noise-propagation strengths $V_r$ of each regulator $r$:

$$E_p = \sum_r R_{pr} V_r + \text{noise}.$$

We assume the noise is Gaussian distributed with unknown variance, and we use a Gaussian prior $P(V_r) \propto e^{-\lambda V_r^2/2}$ on the noise-propagation strengths $V_r$ to avoid over-fitting. The hyper-parameter $\lambda$ is chosen using a cross-validation, fitting the $V_r$ on a random fraction of 80% of the promoters, and maximizing the quality of the predictions on the remaining 20% of the promoters. The quality of fit is quantified by the fraction of the variance in noise levels $E_p$ that is explained by the fit. For our dataset, 17.1% of the variance of the overall dataset was explained by the fit.

## Acknowledgements

We thank D Blank for performing the Western blots, D Bumann for discussions, B Claudi and J Zankl for assistance with flow cytometry, and S Abel and I de Jong for assistance with microscopy.

OKS and EvN designed the study. Experiments were designed by OKS and LW and performed by LW. LW, OKS, and EvN analyzed the data, and EvN developed the theoretical model. LW, OKS, and EvN wrote the paper.

## Additional information

### Funding

| Funder | Grant reference | Author |
| --- | --- | --- |
| Schweizerische Nationalfonds zur Förderung der Wissenschaftlichen Forschung | PZ00P3_126617 | Olin K Silander |
| Schweizerische Nationalfonds zur Förderung der Wissenschaftlichen Forschung | 51RT-0_145680 | Erik van Nimwegen |

The funder had no role in study design, data collection and interpretation, or the decision to submit the work for publication.

### Author contributions

LW, Acquisition of data, Analysis and interpretation of data, Drafting or revising the article; OKS, EN, Conception and design, Analysis and interpretation of data, Drafting or revising the article

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

## Appendix 1

### Supplementary methods

#### Estimating the mean and variance of log-fluorescence levels from FACS data

Visual inspection of the distributions of fluorescence intensities for individual cells containing the same promoter construct shows that almost all of these distributions can be well approximated by a log-normal distribution. We thus chose to characterize the distribution of expression levels of each promoter by the mean and variance of log-fluorescence intensities across cells. Visual inspection of the distributions also indicated that, for almost all promoters, there are a small number of measurements with aberrantly high or low values that are likely due to some measurement artefact, and we designed a Bayesian procedure for automatically discounting these aberrant measurements.

For each clone we typically have around $N = 5000$ independent FACS intensities measured. We denote by $x$ the log-intensity (using natural logs) of an individual cell. We first calculate the mean and variance without taking outliers into account, that is

$$\langle x \rangle = \frac{1}{N} \sum_{i=1}^{N} x_i, \tag{3}$$

and

$$\mathrm{var}(x) = \frac{1}{N} \sum_{i=1}^{N} (x_i - \langle x \rangle)^2, \tag{4}$$

where $x_i$ is the log-intensity of cell $i$. We call these the 'original' mean and variance.

Next we take outliers into account. We assume that, of all $N$ measurements, only a fraction $\rho$ are 'correct' measurements, and the other $(1 - \rho)$ are 'outliers', meaning that these are erroneous measurements. We assume that these 'outliers' derive from a uniform distribution that spans the range of measurements $R = (x_{max} - x_{min})$. Finally, we assume that the distribution of 'correct' measurements is approximately Gaussian with (unknown) mean $\mu$ and variance $\sigma^2$. Under these assumptions, the probability of a measurement of log-intensity $x$ is given by

$$P(x|\mu, \sigma^2, \rho) = \frac{\rho}{\sqrt{2\pi}\sigma} \exp\left[-\frac{(x-\mu)^2}{2\sigma^2}\right] + \frac{1-\rho}{R}. \tag{5}$$

The probability of the entire dataset for a clone is then simply given by

$$P(D|\mu, \sigma^2, \rho) = \prod_{i=1}^{N} P(x_i|\mu, \sigma^2, \rho). \tag{6}$$

We then maximize this probability with respect to $\mu$, $\sigma^2$, and $\rho$. This can be easily done using Expectation-Maximization. The resulting mean $\mu$ and variance $\sigma^2$ are corrected for outliers.

#### Inferring the relation between FACS intensity and GFP molecules per cell

To infer the relationship between FACS intensity per cell and GFP molecules per cell we used quantitative Westerns. For each of eight strains of known FACS intensities, we extracted the protein contents from a fixed number of cells and quantified total GFP intensity. In the same experiment the GFP intensities were measured for known amounts of GFP ranging from 10 to 100 ng. We performed three replicate experiments. In each replicate we measured the GFP intensity of the eight strains, as well as 'reference' intensities of bands loaded with 10, 25, 50, 75,

and 100 ng of GFP. We measured intensities from these gels using both 10 s and 20 s exposure times, giving a total of six replicate measurements of the reference amounts and the eight strains.

*Appendix 1—figure 1* shows the measured GFP intensities *I* as a function of the amount of GFP *w* (weight in grams) for the reference bands in each of the six replicate experiments. Note that there are five points, corresponding to weights of 10, 25, 50, 75, and 100 ng in each curve.

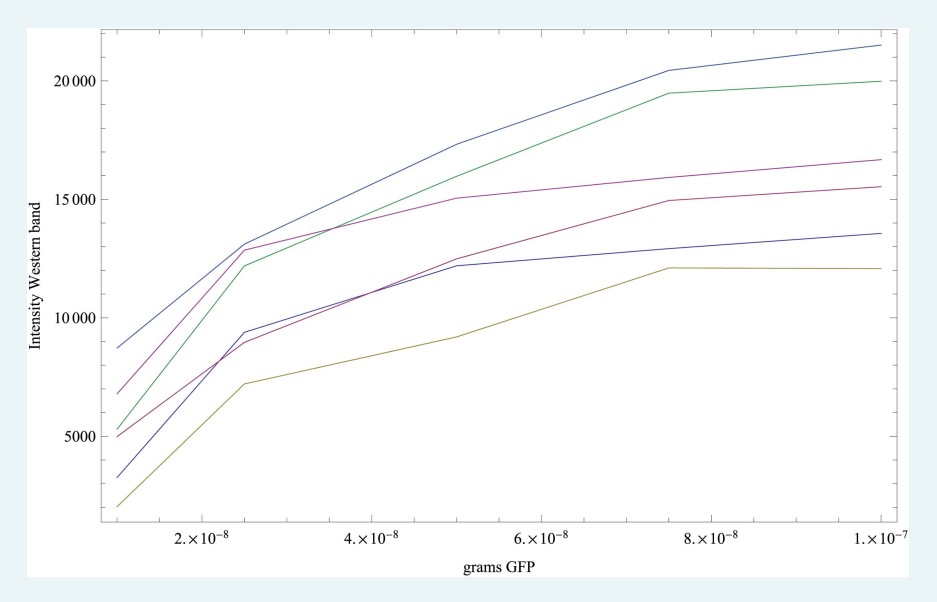

**Appendix 1—figure 1**. Measured intensities of the GFP reference bands as a function of the amount of GFP (in grams) loaded on each band. Each curve corresponds to one replicate (shown in a separate color), and each curve has five data points.

The curves show that the measured intensities are saturating as the amount of GFP increases. Second, the intensity scale varies significantly from replicate to replicate. The simplest linear relationship between *I* and *w* that includes saturation is of the form

$$I = I_{max} \frac{w}{w + w_0}, \tag{7}$$

and inspection of the curves shows that each of them can be reasonably well fitted to this functional form. To infer the amount of GFP corresponding for a particular strain in a particular replicate, we need to infer *w* as a function of the measured value *I*. We thus invert the relationship and find the general form

$$w = w_0 \frac{I}{I_{max} - I}. \tag{8}$$

In other words, our functional form assumes that, for a suitably chosen value $I_{max}$, the weight *w* becomes directly proportional to the transformed variable $I/(I_{max} - I)$. As an example, *Appendix 1—figure 2* shows that, for the first replicate, when plotting *w* as a function of $I/(15631 - I)$, that is, with a value of $I_{max} = 15{,}631$, we obtain an approximately linear relationship. Similar approximately linear relationships are observed for the other replicates as well.

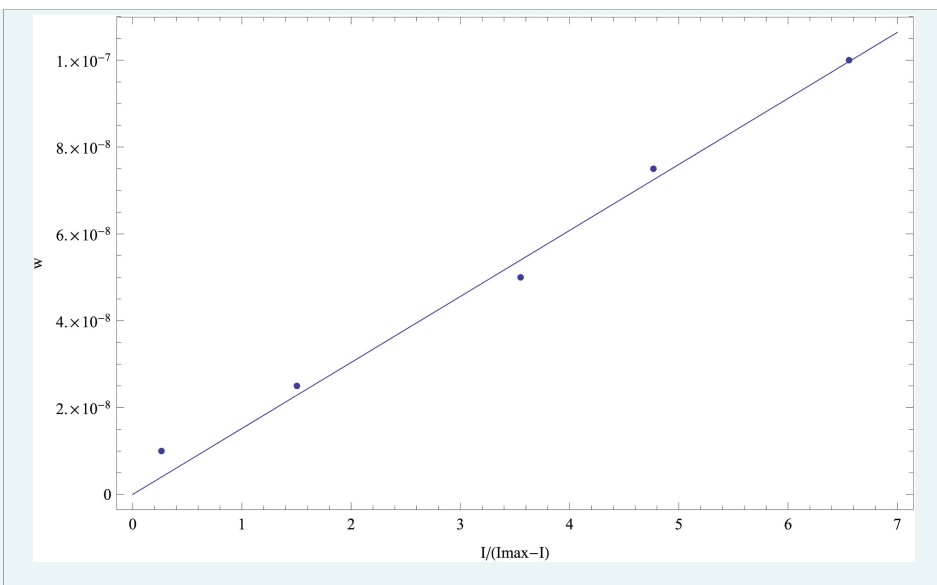

**Appendix 1—figure 2**. For the first replicate, we inferred a saturation value $I_{max} = 15{,}631$. Plotting $w$ as a function of $I/(I_{max} - I)$ we obtain an approximately linear relationship that also approximately goes through the origin $(0, 0)$ (as it should).

To fit $w$ as a function of $I$ for each replicate, we assume that the difference between $w$ and $I/(I_{max} - I)$ is Gaussian distributed with unknown variance $\sigma^2$, that is, for each data point $i$ in a replicate, the weight $w_i$ and its intensity $I_i$ are related through

$$w_i = \epsilon_i + w_0 \frac{I_i}{I_{max} - I_i},\qquad(9)$$

with $\epsilon_i$ the noise, which is Gaussian distributed with unknown variance $\sigma^2$, that is,

$$P(\epsilon|\sigma) = \frac{1}{\sqrt{2\pi}\sigma}\exp\left[-\frac{\epsilon^2}{2\sigma^2}\right].\qquad(10)$$

Using this, the probability of the observed data in a titration curve, given parameters $I_{max}$, $w_0$, and $\sigma$, is:

$$P(\{w\}|\{I\}, I_{max}, w_0, \sigma) \propto \frac{1}{\sigma^n}\exp\left[-\frac{1}{2\sigma^2}\sum_{i=1}^{n}\left(w_i - w_0\frac{I_i}{I_{max} - I_i}\right)^2\right],\qquad(11)$$

where $n = 5$ is the number of points in a titration curve, and we have ignored factors of $\sqrt{2\pi}$ for convenience.

Imagine that we augment our dataset $(\{w\}, \{I\})$ with a single data point $(w_s, I_s)$, where $I_s$ is the measured intensity of strain $s$ and $w_s$ is a hypothesized amount of GFP for this strain. The probability of this entire dataset is given by

$$P(\{w\}, w_s|\{I\}, I_s, I_{max}, w_0, \sigma) = \frac{1}{\sigma^{n+1}}\exp\left[-\frac{1}{2\sigma^2}\sum_{i=1}^{n}\left(w_i - w_0\frac{I_i}{I_{max} - I_i}\right)^2 + \left(w_s - w_0\frac{I_s}{I_{max} - I_s}\right)^2\right],\qquad(12)$$

Formally, we now need to specify a prior $P(I_{max}, w_0, \sigma)$ and integrate over these unknown parameters. We will use a uniform prior over $I_{max}$ and $w_0$ and a scale prior $1/\sigma$ for $\sigma$, that is, formally we want to calculate

$$P(\{w\}, w_s | \{l\}, l_s) = \int dl_{max} dw_0 \frac{d\sigma}{\sigma} P(\{w\}, w_s | \{l\}, l_s, l_{max}, w_0, \sigma). \tag{13}$$

Note, if we additionally integrate over $w_s$ we obtain

$$P(\{w\} | \{l\}, l_s) = \int dw_s P(\{w\}, w_s | \{l\}, l_s), \tag{14}$$

and dividing by this we obtain the posterior distribution of $w_s$:

$$P(w_s | \{w\}, \{l\}, l_s) = \frac{P(\{w\}, w_s | \{l\}, l_s)}{P(\{w\} | \{l\}, l_s)}. \tag{15}$$

To perform the integrals in **Equation 13**, we first simplify the notation by denoting the new data point $(w_s, l_s)$ as $(w_{n+1}, l_{n+1})$, that is, as if it was the $(n+1)$ st data point. The integrand now takes the form

$$P(\{w\} | \{l\}, l_{max}, w_0, \sigma) = \frac{1}{\sigma^{n+1}} \exp\left[ -\frac{1}{2\sigma^2} \sum_{i=1}^{n+1} \left( w_i - w_0 \frac{l_i}{l_{max} - l_i} \right)^2 \right]. \tag{16}$$

To further simplify the notation we write $y_i = l_i / (l_{max} - l_i)$, keeping in mind that the values of $y$ depend on $l_{max}$. Further, for any quantity $x$ that takes on values $x_i$ over the five titration points and the added point, we write averages like

$$\langle x^2 \rangle = \frac{1}{n+1} \sum_{i=1}^{n+1} (x_i)^2, \tag{17}$$

and so on. The integrand can then be rewritten as

$$P(\{w\} | \{l\}, l_{max}, w_0, \sigma) = \frac{1}{\sigma^{n+1}} \exp\left( -\frac{n+1}{2\sigma^2} \left[ \langle w^2 \rangle - 2 w_0 \langle wy \rangle + w_0^2 \langle y^2 \rangle \right] \right). \tag{18}$$

Performing the integral over $w_0$ we obtain

$$P(\{w\} | \{l\}, l_{max}, \sigma) = \frac{1}{\sigma^n \sqrt{\langle y^2 \rangle}} \exp\left[ -\frac{(n+1)\langle w^2 \rangle}{2\sigma^2} \left( 1 - \frac{\langle wy \rangle^2}{\langle w^2 \rangle \langle y^2 \rangle} \right) \right], \tag{19}$$

where we have again ignored prefactors that cancel in the final posterior for $w_s$, that is, **Equation 15**. We next integrate over $\sigma$. Performing this integral we obtain

$$P(\{w\} | \{l\}, l_{max}) = \left( 1 - \frac{\langle wy \rangle^2}{\langle w^2 \rangle \langle y^2 \rangle} \right)^{-n/2} \langle y^2 \rangle^{-1/2}. \tag{20}$$

Notice that the key expression in parentheses is simply one minus the squared correlation coefficient between the variables $w$ and $y$, that is,

$$r^2(y, w) = \frac{\langle wy \rangle^2}{\langle w^2 \rangle \langle y^2 \rangle}. \tag{21}$$

In other words, the values of $w_s$ and $l_{max}$ that maximize the probability are those such that the linear correlation between the resulting values of $y$ and the $w$ values is maximal.

We next need to perform the integral over $l_{max}$. Since this integral cannot be performed analytically, we performed the integrals over $l_{max}$ numerically, separately for each strain $s$ and each of the six replicates. Finally, for each replicate $r$ and strain $s$, we determined the values $w_{rs}$

that have maximal posterior probability. These are our estimated GFP amounts (in grams) for each strain and replicate (**Appendix 1—figure 3**).

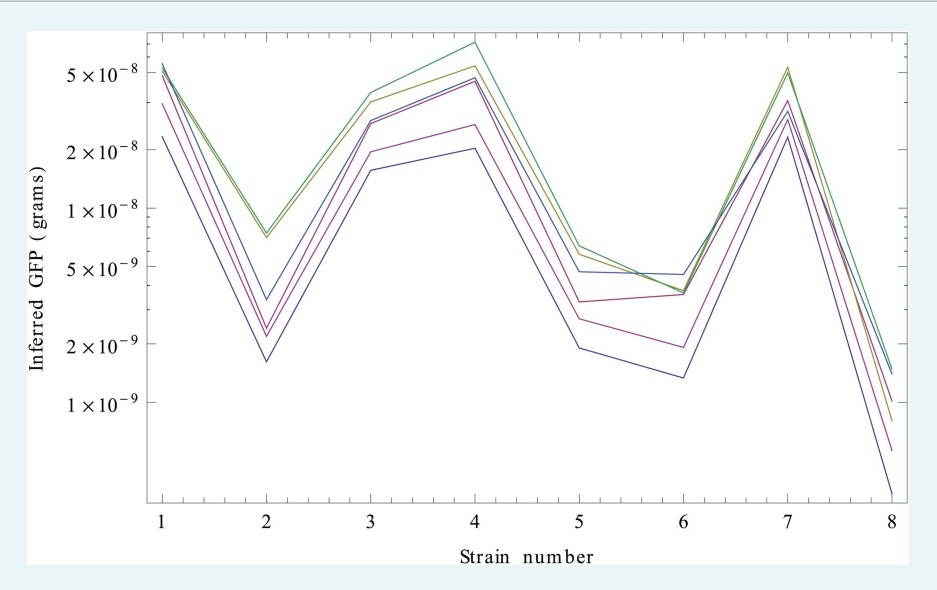

**Appendix 1—figure 3**. Inferred GFP amounts (in grams, vertical axis) for the eight strains (strain numbers shown along the horizontal axis) using the reference data from each replicate. Each color corresponds to a replicate. The vertical axis is shown on a logarithmic scale.

Although the inference clearly separates the high expressed from the low expressed clones, curves from the different replicates seem to be separated by constant shifts from each other. Since the vertical axis is shown on a logarithmic scale, this means that the curves differ by common multiplicative factors. This difference in scale is almost certainly due to an experimental artefact and we will thus normalize for it.

Let $w_s(i)$ be the inferred amount of strain $s$ in replicate $i$. To account for the variability in overall scale, we normalize the inferred log-weights in each replicate by calculating the average log-weight in the replicate, that is,

$$\mu_i = \frac{1}{8} \sum_{s=1}^{8} \log[w_s(i)], \tag{22}$$

and a total average scale of the replicates

$$\mu = \frac{1}{6} \sum_{i=1}^{6} \mu_i, \tag{23}$$

and then transforming the estimated shifts as follows:

$$w_s(i) \rightarrow \tilde{w}_s(i) = w_s(i) e^{\mu - \mu_i}. \tag{24}$$

In addition, dividing the weight $\tilde{w}_s(i)$ by the known weight of a single GFP molecule ($4.482 \times 10^{-20}$ g), we get an estimate of the number of GFP molecules in the bands for each strain. Finally, we used OD measurements to estimate the number of cells loaded on each band, and divided by these to obtain an estimate of the number of GFP molecules per cell for each of the strains across each of the replicates. **Appendix 1—figure 4** shows the inferred GFP molecules per cell for each strain after normalization, which indeed show much less variation across the replicates.

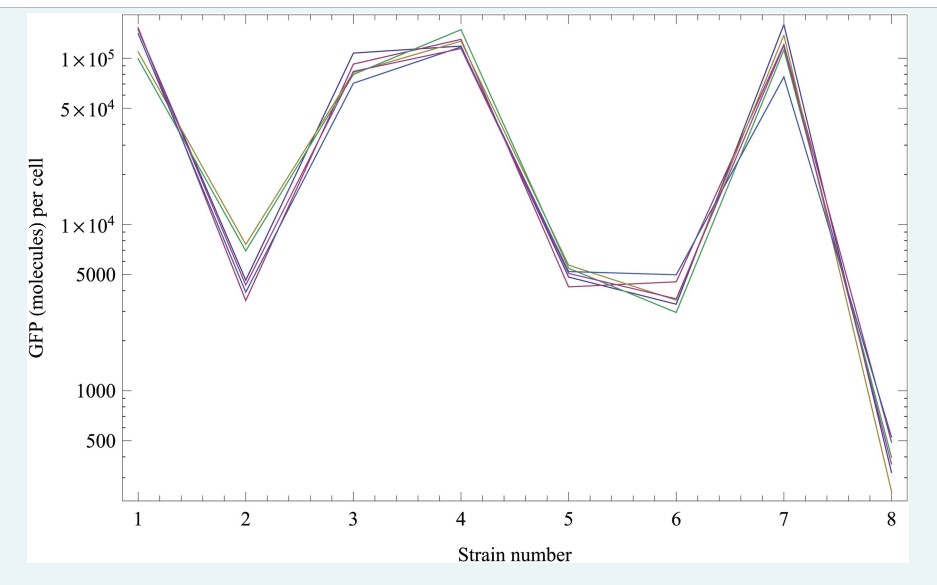

**Appendix 1—figure 4**. Normalized inferred GFP amounts (molecules per cell, vertical axis) for the eight strains (strain numbers shown along the horizontal axis) using the reference data from each replicate. Each color corresponds to a replicate. The vertical axis is shown on a logarithmic scale.

Finally, we compared the inferred GFP amounts for each strain with the FACS intensities measured for that strain. Observing that the variation in both estimated FACS intensities and GFP molecules per cell increases with the mean, it is most natural to compare GFP and FACS levels on a logarithmic scale. Let $f_s$ denote the true log-FACS intensity and $g_s$ denote the true log-GFP molecules per cell. Assuming that GFP molecules per cell and (background-corrected) FACS intensity are directly proportional to each other, the log-levels are related through

$$g_s = f_s + c,$$ (25)

with $c$ a constant. For each strain $s$, we calculated the mean log-FACS intensity $\langle f_s \rangle$ and its variance $\mathrm{var}(f_s)$ across replicate FACS, as well as the mean log-GFP molecules per cell $\langle g_s \rangle$ and its variance $\mathrm{var}(g_s)$ across the quantitative Westerns as described above. Assuming Gaussian deviations between the true and observed levels, the probability of the data given $c$ is given by

$$P(D|c) \propto \prod_s \exp\left[ -\frac{\left(\langle g_s \rangle - \langle f_s \rangle - c\right)^2}{2\left(\mathrm{var}(f_s) + \mathrm{var}(g_s)\right)} \right].$$ (26)

We thus find for the optimal value of $c$:

$$c_* = \sum_s \frac{\langle g_s \rangle - \langle f_s \rangle}{\mathrm{var}(f_s) + \mathrm{var}(g_s)} \left[ \sum_s \frac{1}{\mathrm{var}(f_s) + \mathrm{var}(g_s)} \right]^{-1}.$$ (27)

*Figure 1—figure supplement 2* shows the estimated log-FACS and log-GFP levels including their error bars, together with the optimal fit $c_* = 1.06$.

Consequently, if $F$ is the FACS intensity of a strain (non-log), then the estimated number of GFP molecules per cell $G$ is equal to $G = e^{1.06} F = 2.88F$. Note that, with these estimates, the highest expressed strain, with an average FACS intensity of 37,500, would have about 108,000 molecules of GFP per cell. The lowest expressed strain (with FACS intensity 143) would have 415 molecules per cell. From now on, we will multiply all FACS intensities by 2.88 so that a FACS intensity of $I$ automatically corresponds to the fluorescence of $I$ GFP

molecules, that is, we express FACS fluorescence intensities in units of GFP molecules per cell.

## Comparing mRNA and protein levels

For 94 clones, we quantified mRNA levels using qPCR. The qPCR procedure uses a standard reference curve which allows it to infer the absolute number of molecules of the mRNA of interest in the input sample. Each input sample is created by extracting RNA from a certain number of cells (which we can estimate approximately), and reverse transcribing this RNA into cDNA. Unfortunately, both the total amount of cells used, as well as the efficiency of the reverse transcription, can fluctuate significantly outside of our control, and this will make the total number of molecules detected fluctuate as well. To control for this, we always quantify the absolute number of molecules of two types of mRNAs in parallel for each sample: the mRNAs of the gene of interest and the mRNAs of a reference gene which we are confident is constantly expressed. The reference gene we used was *ihfB*.

For each promoter of interest $p$, we obtained measured mRNA molecule numbers together with mRNA molecule numbers for the reference gene in three separate biological replicates and in three technical replicates for each biological replicate, that is, nine pairs of measurements in total. We denote the log-quantity of the mRNA of promoter $p$ in biological replicate $r$ and technical replicate $i$ as $x_{pri}$, and the log-quantity of the reference gene in the same replicate as $y_{pri}$ (note that this depends on the promoter $p$ because these quantities come from a common sample). To estimate a single log-ratio $x_p - y_p$ between the expression of the gene of interest and the reference gene, we will proceed as follows. First, we will integrate the data from the technical replicates to obtain biological replicate expression $x_{pr}$ and $y_{pr}$. We then combine the differences $d_{pr} = x_{pr} - y_{pr}$ across the biological replicates to obtain the final $d_p = x_p - y_p$.

The statistical model that we use assumes that the difference between the value $x_{pri}$ measured in technical replicate $i$ and the true expression $x_{pr}$ is Gaussian distributed with mean zero and an unknown variance $\sigma_r^2$. Note that we assume that this 'noise' is the same for all promoters $p$, but may fluctuate between biological replicates. We similarly assume the difference between $y_{pri}$ and $y_{pr}$ is Gaussian distributed with variance $\tilde{\sigma}_r^2$. We noted that there is a small fraction of measurements that deviate by large amounts from the measurements in other replicates. We assume that there is a small fraction of measurements that failed in some way, giving erroneous measurement values. To take this into account we will use a mixture model that assumes a small fraction of the measurements come from a uniform distribution that spans the observed range of the data.

Let $R_r = \max_{p,i}(x_{pri}) - \min_{p,i}(x_{pri})$ denote the range of observed values in biological replicate $r$, and let $\rho_r$ denote the fraction of measurements in replicate $r$ that are meaningful, that is, not erroneous. The probability of a single measurement $x_{pri}$ given $x_{pr}$, the variance $\sigma_r^2$ and fraction $\rho_r$ is given by

$$P\left(x_{pri}\middle|x_{pr},\sigma_r^2,\rho_r\right)=\frac{\rho_r}{\sqrt{2\pi}\sigma_r}\exp\left[-\frac{1}{2}\frac{\left(x_{pri}-x_{pr}\right)^2}{2\sigma_r^2}\right]+\frac{1-\rho_r}{R_r}. \tag{28}$$

The probability of all technical replicates for all promoters is then simply given by the product over all promoters $p$ and technical replicates $i$:

$$P\left(\{x_{pri}\}\middle|\{x_{pr}\},\sigma_r^2,\rho_r\right)=\prod_{p,i}P\left(x_{pri}\middle|x_{pr},\sigma_r^2,\rho_r\right). \tag{29}$$

We next maximize this likelihood with respect to the fraction $\rho_r$, the variance $\sigma_r^2$, and the expression levels $x_{pr}$ for all promoters $p$. This optimization can be done using a straightforward Expectation-Maximization scheme.

## Expectation-Maximization

Given a current estimate of $x_{pr}$ of the variance $\sigma_r^2$ and the fraction $\rho_r$, the posterior probability that the technical replicate with value $x_{pri}$ was a meaningful measurement is given by

$$p\left(i\middle|x_{pri}, x_{pr}, \sigma_r^2, \rho_r\right) = \frac{\frac{\rho_r}{\sqrt{2\pi}\sigma_r} \exp\left[-\frac{\left(x_{pri} - x_{pr}\right)^2}{2\sigma_r^2}\right]}{\frac{\rho_r}{\sqrt{2\pi}\sigma_r} \exp\left[-\frac{\left(x_{pri} - x_{pr}\right)^2}{2\sigma_r^2}\right] + \frac{1-\rho_r}{R_r}}. \tag{30}$$

Using these posteriors, the updated value $x'_{pr}$ is given by the mean of the technical replicate measurements, weighted by their posteriors

$$x'_{pr} = \frac{\sum_i x_{pri} p\left(i\middle|x_{pri}, x_{pr}, \sigma_r^2, \rho_r\right)}{\sum_i p\left(i\middle|x_{pri}, x_{pr}, \sigma_r^2, \rho_r\right)}. \tag{31}$$

Given current values of $\rho_r$ and $\sigma_r^2$, we use these equations to iteratively update all $x_{pr}$ until they converge. We then update the values of $\rho_r$ and $\sigma_r^2$ using the following equation

$$\rho'_r = \frac{\sum_{p,i} p\left(i\middle|x_{pri}, x_{pr}, \sigma_r^2, \rho_r\right)}{\sum_{p,i} 1}, \tag{32}$$

that is, the updated $\rho'_r$ is the average of the current posteriors over all promoters and technical replicates. The update equation for the variance is given by

$$\sigma_r'^2 = \frac{\sum_{p,i} \left(x_{pri} - x_{pr}\right)^2 p\left(i\middle|x_{pri}, x_{pr}, \sigma_r^2, \rho_r\right)}{\sum_{p,i} p\left(i\middle|x_{pri}, x_{pr}, \sigma_r^2, \rho_r\right)}. \tag{33}$$

After each update of $\sigma_r^2$ and $\rho_r$, all $x_{pr}$ are updated until convergence again, and this is iterated until the $\sigma_r^2$ and $\rho_r$ converge. Exactly analogous Expectation-Maximization equations are used to optimize the values $\tilde{\sigma}_r$, $\tilde{\rho}_r$, and all $y_{pr}$ of the reference gene measurements.

*Appendix 1—Table 1* shows the fitted fractions and variances for each of the replicates.
**Appendix 1—Table 1**. Fitted variances and fractions of meaningful measurements for the genes of interest ($\sigma^2$, $\rho$) as well as for the reference gene measurements ($\tilde{\sigma}^2$, $\tilde{\rho}$) for each of the three biological replicates

| Replicate | $\sigma^2$ | $\rho$ | $\tilde{\sigma}^2$ | $\tilde{\rho}$ |
|---|---|---|---|---|
| 1 | 0.0252 | 1.0 | 0.0116 | 0.934 |
| 2 | 0.0113 | 0.981 | 0.0118 | 0.988 |
| 3 | 0.0329 | 0.956 | 0.0072 | 0.955 |

We see that, for the majority of replicates, the noise level lies around 0.01 (meaning a measurement error bar of about 0.1 on log-expression), but it is two and threefold higher for measurements of the genes of interest in two replicates. The fraction of correct measurements ranges from about 93% to almost 100% across the replicates.

When the variances and fractions have been optimized, we obtain the final technical replicate-averaged quantities $x_{pr}$ and $y_{pr}$ and we determine final variances $\sigma_{pr}^2$ and $\tilde{\sigma}_{pr}^2$ for each of these averages. These final variances are calculated as follows. For each promoter $p$ and each biological replicate $r$, we determine the effective number of correct measurements as

$$n_{pr} = \sum_i p\left(i\middle|x_{pri}, x_{pr}, \sigma_r^2, \rho_r\right), \tag{34}$$

and the final variance is then given by

$$\sigma^2_{pr} = \frac{\sigma^2_r}{n_{pr}}. \tag{35}$$

Analogously, for the reference gene measurements we have

$$\tilde{n}_{pr} = \sum_i p\left(i \mid y_{pri}, y_{pr}, \tilde{\sigma}^2_r, \tilde{\rho}_r\right), \tag{36}$$

and the final variance

$$\tilde{\sigma}^2_{pr} = \frac{\tilde{\sigma}^2_r}{\tilde{n}_{pr}}. \tag{37}$$

## Combining the biological replicates

For each promoter $p$, we want to estimate the log-expression ratio $x_p - y_p$ by combining the estimated values $x_{pr}$ and $y_{pr}$ from each of the replicates, taking into account that these values have different variances for different replicates. For a protein $p$ and replicate $r$, the estimated log-expression difference $d_{pr}$ is

$$d_{pr} = x_{pr} - y_{pr}. \tag{38}$$

The variance $\tau^2_{pr}$ associated with that estimated difference is

$$\tau^2_{pr} = \sigma^2_{pr} + \tilde{\sigma}^2_{pr}. \tag{39}$$

Inspection of the variation in $d_{pr}$ across biological replicates, relative to their uncertainties $\tau_{pr}$, makes it clear that, in addition to the uncertainty in each of the estimates $d_{pr}$, there is substantial variation in $d_{pr}$ across the biological replicates, which is quite different for different promoters; that is, for some promoters the biological replicates give very consistent $d_{pr}$, lying within the error bars $\tau_{pr}$, whereas for other promoters the variation in $d_{pr}$ is much larger than the error bars $\tau_{pr}$, indicating that there must be additional variance across replicates.

We will assume that the true value $d^t_{pr}$ is given by the mean $d_p$ for the promoter plus a biological replicate variation $\delta_{pr}$

$$d^t_{pr} = d_p + \delta_{pr}, \tag{40}$$

and we will assume that the deviation $\delta_{pr}$ is Gaussian distributed with mean zero and unknown variance $\tau^2_p$. The probability of the estimate $d_{pr}$ given its variance $\tau^2_{pr}$, the true value $d_p$, and the biological replicate variance $\tau^2_p$ is given by

$$P\left(d_{pr} \mid d_p, \tau^2_{pr}, \tau^2_p\right) = \frac{1}{\sqrt{2\pi\left(\tau^2_{pr} + \tau^2_p\right)}} \exp\left[-\frac{\left(d_{pr} - d_p\right)^2}{2\left(\tau^2_{pr} + \tau^2_p\right)}\right]. \tag{41}$$

The probability of the data combining all biological replicates for the promoter is simply

$$P\left(d_{p1}, d_{p2}, d_{p3} \mid d_p, \tau^2_p, \tau^2_{p1}, \tau^2_{p2}, \tau^2_{p3}\right) = \prod_{r=1}^{3} P\left(d_{pr} \mid d_p, \tau^2_{pr}, \tau^2_p\right). \tag{42}$$

For each promoter $p$, we now maximize this probability with respect to both $\tau^2_p$ and $d_p$. Given a fixed value of $\tau^2_p$, the optimal value of $d_p$ is given by the weighted sum

$$d_p = \frac{\sum_r \frac{d_{pr}}{\tau_p^2 + \tau_{pr}^2}}{\sum_r \frac{1}{\tau_p^2 + \tau_{pr}^2}}. \tag{43}$$

Substituting this optimal $d_p$ into the probability (**Equation 42**), the expression becomes a function of the variance $\tau_p^2$ only, and we numerically determine the optimal value of $\tau_p^2$ for each $p$. In this way we obtain a final estimate $d_p$ for each promoter. The variance $\sigma_p^2$ associated with this final estimate is given by

$$\sigma_p^2 = \left[ \sum_r \frac{1}{\tau_p^2 + \tau_{pr}^2} \right]^{-1}. \tag{44}$$

**Figure 1—figure supplement 3** shows the relationship between protein levels (estimated by FACS) and estimated mRNA levels for the 94 strains for which we measured mRNA levels using qPCR. We see there is a very good correlation between protein and mRNA levels (Pearson correlation $r^2 \approx 0.82$). Note that, for a given promoter, the average protein level $p$ is related to the average mRNA level $m$ by the ratio of the translation rate $\lambda$ and protein decay rate $\mu$, that is,

$$p = \frac{\lambda}{\mu} m. \tag{45}$$

Since GFP is very stable compared to the duplication rate of our cells, for our system the protein decay rate $\mu$ is approximately equal to the growth rate of the cells, and thus constant across the promoters. Consequently, the fact that 82% of the variation in protein levels is explained by variations in mRNA levels suggests that the translation rate $\lambda$ shows relatively small variations across the strains. Below we use this data to more rigorously estimate variation in translation rates across the strains.

## Estimating relative translation rates

As before, we denote by $d_p$ the relative (to *ihfB*) log-mRNA level of promoter $p$, and we will denote by $y_p$ the log-protein number per cell (as measured by FACS) for promoter $p$. Denoting by $m$ the absolute number of mRNAs per cell for the reference gene *ihf*, by $\lambda$ the average translation rate, and by $\mu$ the protein decay rate (as a consequence of cell growth), $y_p$ and $d_p$ are related through

$$e^{y_p} = \frac{\lambda e^{\delta_p}}{\mu} e^{d_p} m, \tag{46}$$

where we have written the translation rate $\lambda_p$ of promoter $p$ in terms of the average translation rate $\lambda$, and a promoter specific deviation $\delta_p$, that is, $\lambda_p = \lambda e^{\delta_p}$. Defining $e^c = \lambda m / \mu$ we have

$$y_p = d_p + c + \delta_p. \tag{47}$$

Using that our estimate of $d_p$ is Gaussian distributed with standard deviation $\sigma_p$ and assuming that $\delta_p$ is Gaussian distributed with mean 0 and standard deviation $\tau$, the probability of our data given the $\sigma_p$, $c$, and $\tau$ is

$$P\left( \{d_p, y_p, \delta_p\} \middle| c, \{\sigma_p\}, \tau \right) = \prod_p \frac{1}{2\pi\sigma_p\tau} \exp\left[ -\frac{1}{2} \left( \frac{y_p - d_p - c - \delta_p}{\sigma_p} \right)^2 - \frac{\delta_p^2}{2\tau^2} \right]. \tag{48}$$

To estimate the variance $\tau^2$ we integrate over all $\delta_p$ and $c$ (using a uniform prior). To simplify the notation of the result we write $w_p = 1/(\sigma_p^2 + \tau^2)$ and $\Delta_p = y_p - d_p$. We then have

$$P\left(\{d_p, y_p\}\middle|\{\sigma_p\}, \tau\right) \propto \frac{\Pi_p \sqrt{w_p}}{\sqrt{\sum_p w_p}} \exp\left[-\frac{1}{2}\left(\sum_p w_p \Delta_p^2 - \frac{\left(\sum_p w_p \Delta_p\right)^2}{\sum_p w_p}\right)\right].$$ (49)

We numerically determine the value of $\tau$ that maximizes this likelihood and find $\tau_* = 0.47$. Using this maximum likelihood value of $\tau$, the maximum likelihood value of $c$ is given by

$$c_* = \frac{\sum_p w_p \Delta_p}{\sum_p w_p} = 7.06.$$ (50)

The fit $y = c + d$ is shown as the black line in **Figure 1—figure supplement 3**.

Finally, using $\tau_*$ and $c_*$, we determine the most likely values of the $\delta_p$. We find

$$\delta_p = \frac{\Delta_p - c_*}{1 + \sigma_p^2 / \tau_*^2},$$ (51)

with a standard deviation of

$$\sigma(\delta_p) = \left(\frac{1}{\tau_*^2} + \frac{1}{\sigma_p^2}\right)^{-1/2}.$$ (52)

**Appendix 1—figure 5** shows the estimated values of $\delta_p$, together with their error bars $\sigma(\delta_p)$, as a function of the log-protein level $y_p$. We see that, for the large majority of promoters, the estimated translation rate is within 2–3-fold of the average translation rate (i.e., $|\delta_p| < 1$), confirming that there is relatively little variation in translation rates. For the most extreme example, the translation rate is approximately $e^{1.9} = 6.6$ fold lower than the average translation rate.

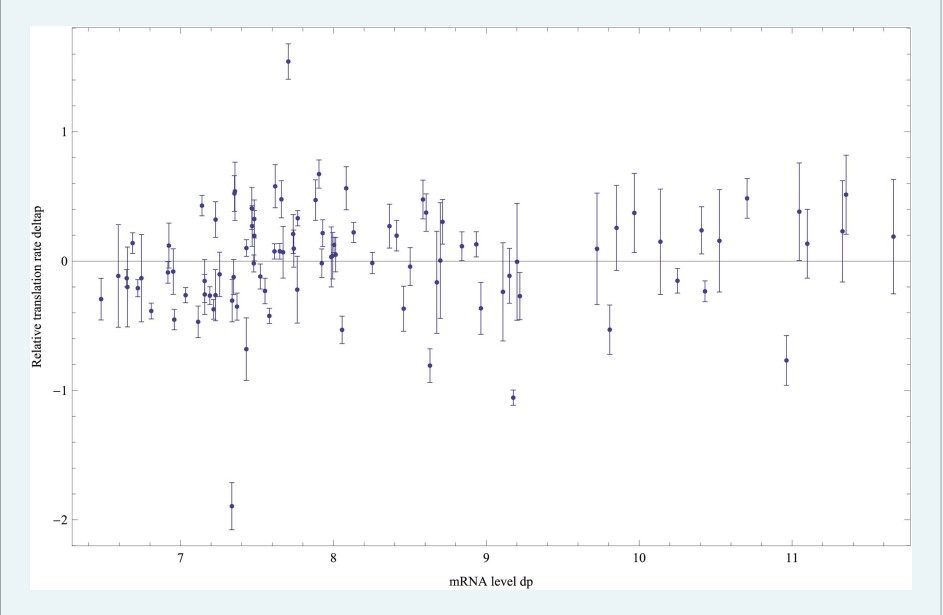

**Appendix 1—figure 5**. Estimated relative log-translation rates $\delta_p$ and their error bars $\sigma(\delta_p)$ (vertical axis) as a function of the log-mRNA level relative to *ihfB*, $d_P$, for each promoter $p$.

The figure also shows that there is no correlation between the relative translation rate $\delta_p$ and the log-mRNA level $d_p$. We also find no correlation of $\delta_p$ with either log-protein level $y_p$ or the variance of the log-protein level (data not shown). Thus, in summary, these results show that translation rates vary relatively little across the constructs and do not systematically scale with either protein or mRNA levels.

## Minimal expression noise as a function of mean expression

To model the noise distribution of our promoters we start with the simple case in which there are constant rates of transcription, translation, mRNA decay, and protein decay. Let $\lambda_m$ be the rate of transcription per unit time, $\mu_m$ the rate of mRNA decay (per mRNA per unit time), $\lambda_p$ the rate of translation (per mRNA per unit time), and $\mu_p$ the rate of protein decay (per protein per unit time). Note that in our case all proteins decay at the same rate and, because the decay rate of GFP is relatively small compared to the dilution rate as a consequence of cell growth, the rate $\mu_p$ is effectively given by the growth rate of the cells.

In *Shahrezaei and Swain (2008)* an analytical expression was derived for the distribution $P(n|\lambda_m, \mu_m, \lambda_p, \mu_p)$ under the assumption that the rate of protein decay is small compared with the rate of mRNA decay. In *E. coli* the typical mRNA decay rate is of the order of 5 min (*Bernstein et al., 2002*). In the minimal media with glucose in which our cells are grown, the doubling time is more than 30 min, so that the protein decay is indeed smaller than the mRNA decay rate by a factor of approximately 6. Since this is not a very large factor, one may worry that, for stable mRNAs, the approximation breaks down. Fortunately, in *Shahrezaei and Swain (2008)* it was also shown (by simulation) that, as long as the mRNA decay rate is at least as large as the protein decay rate, then the approximation is still quite accurate. We will thus assume that we can use this approximation.

Under this approximation the stationary distribution of the number of proteins per cell depends only on the following two ratios:

$$a = \frac{\lambda_m}{\mu_p},$$ 

(53)

and

$$b = \frac{\lambda_p}{\mu_m}.$$ 

(54)

The ratio $a$ gives the expected number of transcripts that are produced during the life-time of a single protein, which in our case effectively means the doubling time of the cells, that is, $a$ is the expected number of transcription events per cell cycle. The ratio $b$ gives the expected number of proteins that are produced from a single mRNA during its life-time. This is sometimes referred to as the 'burst size', that is, typically one assumes $b > 1$ and, given that mRNA decay is faster than protein decay, the proteins are produced 'fast' from a single mRNA compared with the life-time of a typical protein, that is, in a burst.

The limit distribution $P(n|a, b)$ is given by a negative binomial

$$P(n|a, b) = \frac{\Gamma(a+n)}{\Gamma(n+1)\Gamma(a)} \left(\frac{b}{b+1}\right)^n \left(1 - \frac{b}{b+1}\right)^a.$$ 

(55)

This distribution has a mean

$$\langle n \rangle = ab,$$ 

(56)

and variance

$$\mathrm{var}(n) = ab(1+b) = \langle n \rangle(1+b).$$ 

(57)

We extend this simple model by assuming the ratios $a$ and $b$ fluctuate themselves (most likely on a somewhat slower time scale). Although we will not attempt to specify the molecular origins of these fluctuations in $a$ and $b$, they likely include fluctuations in the concentrations of polymerases, ribosomes, and TFs that regulate the promoter in question. Such fluctuations would contribute to the extrinsic noise of the promoters, since they would equally affect two copies of the same promoter in the same cell. However, they may also include fluctuations in the state of the promoter itself, and such fluctuations would contribute to the intrinsic noise.

We will assume that the fluctuations in these ratios of rates are multiplicative, that is, proportional to the means $\langle a \rangle$ and $\langle b \rangle$:

$$\mathrm{var}(a) = \langle a \rangle^2 \sigma_a^2, \tag{58}$$

and

$$\mathrm{var}(b) = \langle b \rangle^2 \sigma_b^2. \tag{59}$$

We then find for the total variance of $n$

$$\mathrm{var}(n) = \langle n \rangle^2 \left( \sigma_a^2 + \sigma_b^2 + \sigma_a^2 \sigma_b^2 \right) + \langle n \rangle \left[ 1 + \langle b \rangle \left( 1 + \sigma_b^2 \right) \right]. \tag{60}$$

To simplify the notation, we introduce the variable

$$\sigma_{ab}^2 = \sigma_a^2 + \sigma_b^2 + \sigma_a^2 \sigma_b^2, \tag{61}$$

and the renormalized burst size

$$\beta = 1 + \langle b \rangle \left( 1 + \sigma_b^2 \right). \tag{62}$$

With these definitions we have

$$\mathrm{var}(n) = \langle n \rangle^2 \sigma_{ab}^2 + \beta \langle n \rangle, \tag{63}$$

which brings out most clearly that there is a term proportional to $\langle n \rangle^2$ that results from fluctuations in $a$ and $b$, and a term proportional to $\langle n \rangle$ that results from Poisson fluctuations in mRNA and protein production, and is proportional to the burst size.

We now want to relate this expression to variations in log-fluorescence intensities as measured using FACS. Here it is important to note that the log-fluorescence intensity per cell is the result of a combination of fluorescence coming from GFP proteins and background fluorescence of the cell.

## Background fluorescence

To estimate background fluorescence, we performed three replicate measurements of populations of cells without any plasmid and three replicate measurements of populations of cells containing an empty plasmid (not containing a GFP gene). *Appendix 1—figure 6* shows the reverse cumulative distributions of observed intensities in these control populations (colored lines).

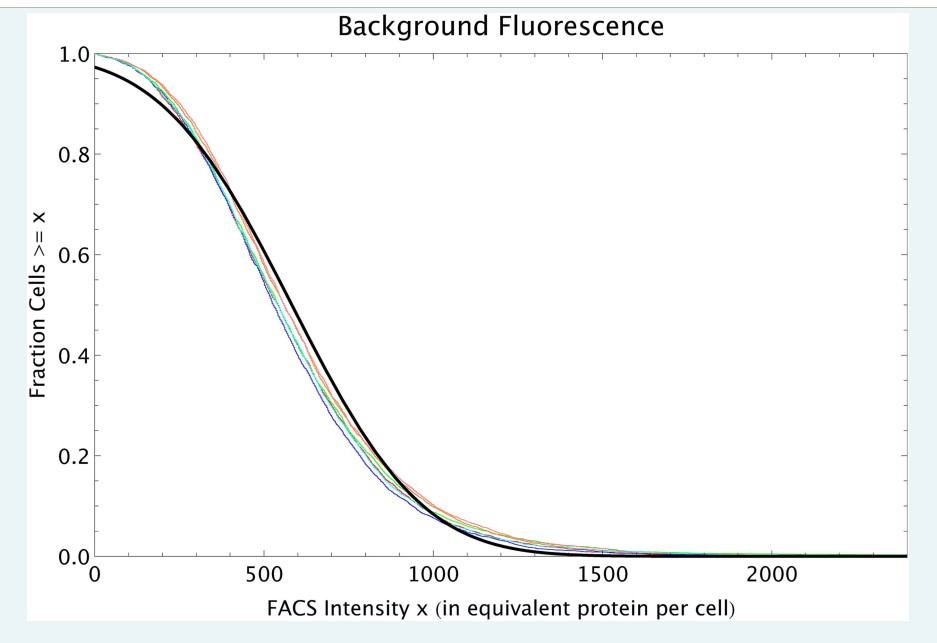

**Appendix 1—figure 6.** Reverse cumulative distributions of the FACS intensities per cell (multiplied by 2.88 so as to correspond to the equivalent of GFP proteins per cell) for MG1655 cells without a plasmid (red, blue and green curves) and MG1655 cells with an empty plasmid (orange, pink and cyan curves). The black line shows a Gaussian distribution with matching mean and variance.

The curves show that each replicate shows a highly similar distribution of fluorescence levels, and pooling the data from all replicates we find a mean background fluorescence of $n_{bg} = 582.3$ with a standard deviation of $\sigma_{bg} = 302.9$. As shown by the black curve in **Appendix 1—figure 6**, the distribution of background fluorescences is reasonably approximated by a Gaussian with the same mean and standard deviation.

## Relating measured variations to the theoretical expression

Let $n_{meas}$ denote the measured FACS intensity of a cell. We will write this measured intensity as the sum of an average background fluorescence $n_{bg}$, the average number of proteins $\langle n \rangle$, and a fluctuation of size $\epsilon\sqrt{\text{var}(n)}$:

$$n_{meas} = n_{bg} + \langle n \rangle + \epsilon\sqrt{\text{var}(n)}, \tag{64}$$

Here $\epsilon$ is a quantity that fluctuates from cell to cell, which has mean zero $\langle \epsilon \rangle = 0$, and variance one, that is, $\langle \epsilon^2 \rangle = 1$.

We will assume that the fluctuations $\epsilon\sqrt{\text{var}(n)}$ are small relative to the mean $\langle n_{meas} \rangle = n_{bg} + \langle n \rangle$. We can then write for the logarithm of the measured FACS intensity

$$\log[n_{meas}] \approx \log[\langle n_{meas} \rangle] + \epsilon\frac{\sqrt{\text{var}(n)}}{\langle n_{meas} \rangle}, \tag{65}$$

and find for the variance in log-scale of the measured FACS intensities

$$\text{var}(\log[n_{meas}]) = \frac{\text{var}(n)}{\langle n_{meas} \rangle^2}. \tag{66}$$

If we substitute the expression (**Equation 63**) for the numerator, we obtain

$$\text{var}(\log[n_{meas}]) = \sigma_{ab}^2\left(1 - \frac{n_{bg}}{\langle n_{meas} \rangle}\right)^2 + \frac{\beta}{\langle n_{meas} \rangle}\left(1 - \frac{n_{bg}}{\langle n_{meas} \rangle}\right). \tag{67}$$

The left panel of **Appendix 1—figure 7** shows the mean and variances of the log-FACS intensities of all native promoters. This scatter shows that, as a function of the mean FACS intensity, there is a sharp lower bound on the observed variances. The red curve shows that this lower bound can be well-fitted by a function of the form (**Equation 67**), where we used parameters $\sigma_{ab}^2 = 0.025$ and $\beta = 450$. Note that the value of $\sigma_{ab}^2$ determines the variance in the limit of large means whereas $\beta$ controls the curvature at lower means. We fitted these two parameters by hand. Their interpretation is that, $\sigma_{ab}^2$ corresponds to the minimal amount of cell-to-cell variation in the product $ab$ that is possible for any promoter architecture. The variable $\beta = 450$ roughly corresponds to the burst size.

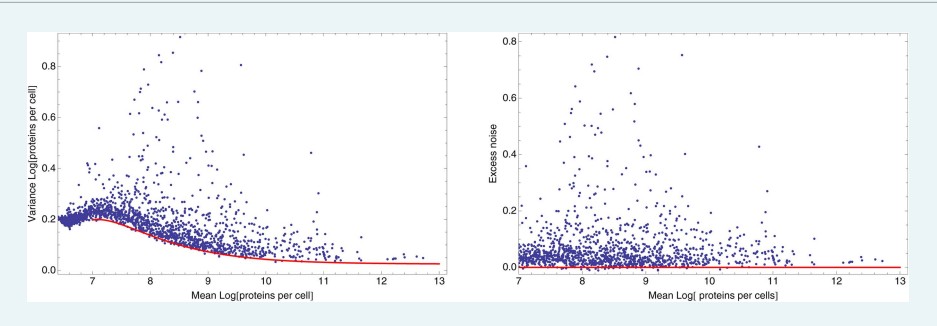

**Appendix 1—figure 7**. Dependence between mean and variance of log FACS intensities. Left panel: Means and variances of log-FACS intensities of all native promoters (blue dots) together with a fitted lower bound on the variance as given by **Equation 67** using $\sigma_{ab}^2 = 0.025$ and $\beta = 450$ (red curve). Right panel: Excess noise (obtained by subtracting the fitted lower bound from the variance) as a function of mean log-FACS intensity for all native promoters (blue dots). The red line shows the x-axis.

Note that the log-fluorescence on the horizontal axis corresponds to the sum of fluorescence resulting from GFP molecules and the background fluorescence. The estimated background level $n_{bg} = 582.3$ corresponds to a log-fluorescence of 6.37. The region on the horizontal axis between 6.37 and $7 \approx \log(2 \times 582.3)$ thus corresponds to cells where the fluorescence due to GFP molecules is less than the background fluorescence. In this regime the noise distribution results from a combination of fluctuations in background fluorescence and in protein numbers (which may be correlated because part of these fluctuations may result from fluctuations in cell size) and our noise model (**Equation 67**) breaks down. In the following we will focus on those promoters with fluorescence due to GFP at least as large as the background fluorescence, that is, with mean log-fluorescence larger than 7.

To obtain a deviation of each promoter's variance from the minimal variance that is possible at its expression level, we define the excess noise $\eta$ as the difference between a promoter's variance and its fitted minimal variance $\sigma_{min}^2(\mu)$ as given by **Equation 67** with $\beta = 450$ and $\sigma_{ab}^2 = 0.025$:

$$\eta = \sigma^2 - \sigma_{min}^2(\mu). \tag{68}$$

The right panel of **Appendix 1—figure 7** shows the excess noise levels of all native promoters as a function of their means. The figure shows that, with this correction, there is no longer any systematic dependence between mean expression levels and noise. Therefore, we can use excess noise as a measure of transcriptional noise that allows us to compare noise levels of promoters with different mean expression levels.

**Appendix 2**

## FACS selection

As explained in the 'Materials and methods', for both the medium and high expression evolutionary runs, the desired expression level $\mu_*$ is taken from the expression level of a reference promoter from the library of *E. coli* promoters. At each selection round we measure the expression $\mu_*$ of the reference promoter, and set the center of the FACS selection window to $\mu_*$. We then set the width of the selection window such that 5% of the cells have expression levels within the selection window.

Although, in principle, the FACS selection should work such that a cell with expression level anywhere within the selection window has 100% probability to be selected, and 0% probability to be selected if the cell's expression is anywhere outside the selection window, it is unrealistic to assume that the boundaries of the selection window are so precisely defined in practice. As illustrated below, comparison of the population's expression levels before and after selection shows that the probability for a cell with log-fluorescence $x$ to be selected can be well-approximated by

$$f(x|\mu_*, \tau) = \exp\left[-\frac{(x-\mu_*)^2}{2\tau^2}\right],\tag{69}$$

where $\mu_*$ is the desired expression level and $\tau$ corresponds to the width of the selection window.

Note that, for a promoter with mean expression $\mu$ and variance $\sigma^2$, the fraction $P(x|\mu, \sigma)$ of its cells that have expression level $x$ is given by

$$P(x|\mu, \sigma)dx = \frac{1}{\sqrt{2\pi}\sigma}\exp\left[-\frac{(x-\mu)^2}{2\sigma^2}\right]dx.\tag{70}$$

Consequently, the 'fitness' of this promoter, that is, the fraction of its cells that are selected in the FACS, is given by

$$f(\mu, \sigma|\mu_*, \tau) = \int dx f(x|\mu_*, \tau)P(x|\mu, \sigma) = \sqrt{\frac{\tau^2}{\tau^2+\sigma^2}}\exp\left[-\frac{(\mu-\mu_*)^2}{2(\tau^2+\sigma^2)}\right].\tag{71}$$

To infer the values of $\mu_*$ and $\tau$ that apply to our evolutionary runs, we performed a number of experiments in which we:
1. Took a population from one of the rounds of our evolutionary runs.
2. Measured its distribution of log-fluorescence levels.
3. Set the selection window $[\mu_* - \delta, \mu_* + \delta]$ such that a percentage $p$ of the population has log-fluorescence levels within this selection window.
4. Performed selection and re-measured the log-fluorescence levels of the selected population.

As shown in **Figure 1B** of the main paper, in the evolutionary runs in which we are selecting for high expression, the selection window is changing at every round of the evolutionary run. In contrast, in the evolutionary runs selecting for medium expression, the selection window is essentially constant from round three through round five. We thus decided to focus on inferring the precise fitness function that acted during these three rounds of selection.

We took the evolved populations from the third and fifth rounds of the evolutionary runs selecting for medium expression and performed another round of selection on them, selecting 5% of the population closest to the desired log-fluorescence $\mu_*$. In addition, we also performed a round of less stringent selection on these populations, selecting 25% closest to the desired level, and a round of more stringent selection, selecting only the 1% of the population closest

to the desired level. Besides measuring the log-fluorescence levels of the population both before and after the round of selection, we also selected dozens of clones from the populations before and after the selection and measured the entire distribution of log-fluorescence levels for these clones. **Appendix 2—figure 1** shows the means and variances of the log-fluorescence distributions of these clones.

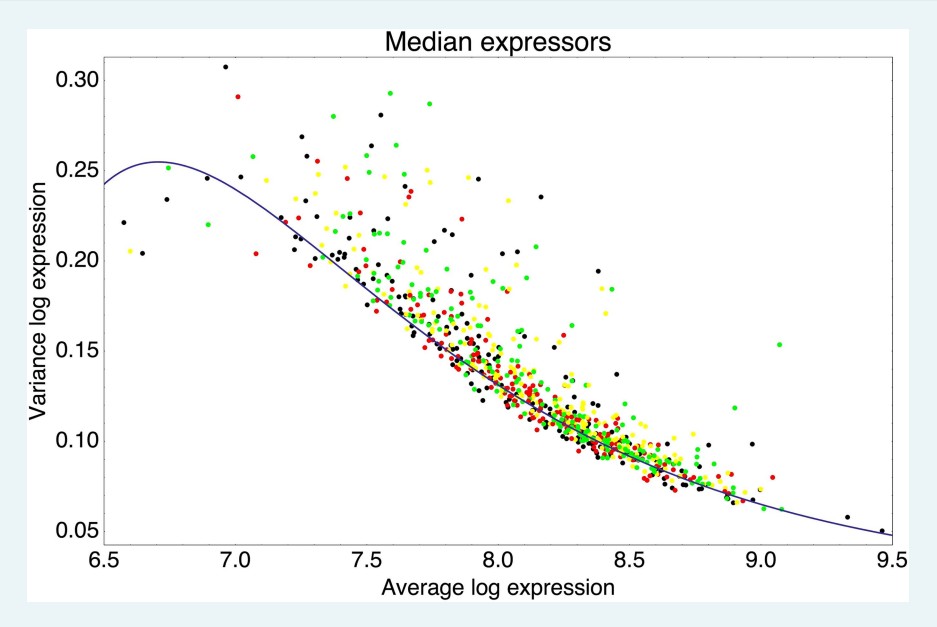

**Appendix 2—figure 1**. Means and variances of the log-fluorescence levels of clones from the third and fifth rounds of the evolutionary runs in which we selected for medium expression (black dots), and clones obtained after performing another round of selection on these populations, selecting either 1% (red), 5% (yellow), or 25% (green) of the population closest to the desired log-fluorescence $\mu_*$. The blue curve shows a fit of the typical variance $\sigma^2$ as a function of the mean $\mu$: $\sigma^2(\mu) = 0.02 + 384e^{-\mu} - 156{,}915e^{-2\mu}$.

Intuitively, one might think that the relative fitness $f(x)$ of each log-fluorescence level $x$ could be easily estimated by simply measuring the ratio of the fraction of the population $p'(x)$ with log-fluorescence level $x$ after selection and the fraction $p(x)$ with log-fluorescence $x$ before selection. However, the single cells that were selected in the FACS each grow into an entire population of cells before the 'after selection' population is measured again. Thus, a selected cell containing a promoter with a given mean $\mu$ and variance $\sigma^2$ will contribute an entire population of cells with this distribution, even though the individual cell may itself have had a log-fluorescence that was in one of the tails of this distribution. Thus, in general the distribution of log-fluorescence levels in the population after selection may be much wider than the actual selection window itself.

Before selection, the population consisted of a mixture containing (unknown) fractions $\rho(\mu, \sigma)$ of cells containing promoters with mean $\mu$ and variance $\sigma^2$. This gives rise to an overall distribution $p(x)$ of expression levels given by

$$p(x) = \int d\mu\, d\sigma \frac{\rho(\mu, \sigma)}{\sqrt{2\pi}\sigma} \exp\left[-\frac{(x-\mu)^2}{2\sigma^2}\right]. \tag{72}$$

Unfortunately we cannot uniquely infer $\rho(\mu, \sigma)$ from knowing only the distribution $p(x)$. However, as shown in **Appendix 2—figure 1**, for the clones in these populations, the large majority of promoters have variances $\sigma^2$ lying in a narrow band as a function of $\mu$. We thus chose to make the approximation that *all* promoters in the population have variances $\sigma^2$ that are uniquely

determined by their mean expression $\mu$, and we used the fit $\sigma^2(\mu)$ shown as the blue curve in **Appendix 2—figure 1**. This simplifies the problem from inferring a two-dimensional distribution $\rho(\mu, \sigma)$ to inferring a one-dimensional distribution $\rho(\mu)$, that is,

$$p(x) = \int d\mu \, \frac{\rho(\mu)}{\sqrt{2\pi}\sigma} \exp\left[-\frac{(x-\mu)^2}{2\sigma^2(\mu)}\right]. \tag{73}$$

Applying selection with target log-fluorescence $\mu_*$ and width $\tau$ to this distribution, we obtain a new distribution

$$\rho'(\mu) = C\rho(\mu)\sqrt{\frac{\tau^2}{\tau^2 + \sigma^2(\mu)}} \exp\left[-\frac{(\mu - \mu_*)^2}{2\left(\tau^2 + \sigma(\mu)^2\right)}\right], \tag{74}$$

where $C$ is a normalization constant that ensures $\int d\mu \rho'(\mu) = 1$. The population distribution of log-fluorescence levels after selection is then

$$p'(x) = \int d\mu \, \frac{\rho'(\mu)}{\sqrt{2\pi}\sigma} \exp\left[-\frac{(x-\mu)^2}{2\sigma^2(\mu)}\right]. \tag{75}$$

To infer the parameters of the fitness function for each selection that we performed, we fit the distribution $\rho(\mu)$ and parameters $\mu_*$ and $\tau$ that lead to an optimal fit to the observed distributions $p(x)$ and $p'(x)$. **Appendix 2—figure 2** shows the inferred and observed distributions, as well as the inferred fitness function, for each of the six selection experiments that we performed.

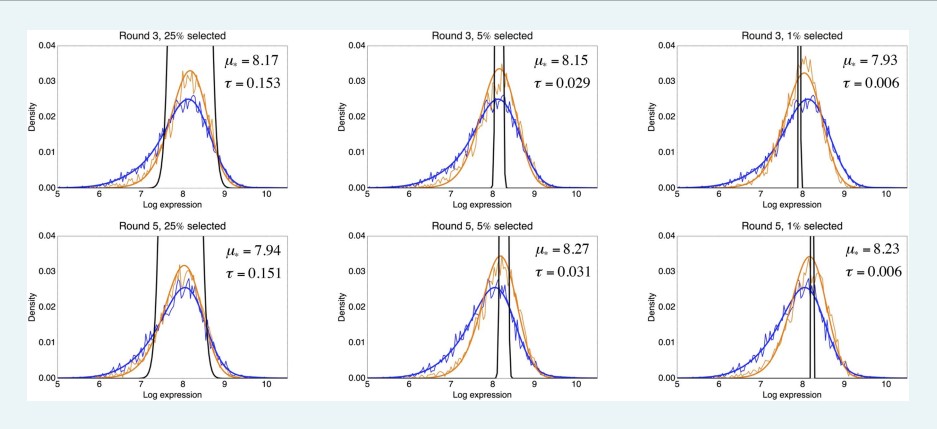

**Appendix 2—figure 2**. Inference of the fitness function from the observed log-fluorescence distribution before and after a round of selection. Each panel corresponds to one selection experiment with the title indicating on which population an extra round of selection was performed, that is, a population either from the third or fifth round of the evolutionary run for medium expression, and what fraction of the population was selected. The thin blue line indicates the observed log-fluorescence distribution $p(x)$ before selection, and the thin orange line the observed distribution $p'(x)$ after selection. The thick lines show the corresponding fitted distributions. The inferred selection window $f(x|\mu_*, \tau)$, that is, **Equation 69**, is indicated in black, and its parameters $\mu_*$ and $\tau$ are indicated in each panel as well.

The figure shows that the distributions $p(x)$ and $p'(x)$ can be well fit by this model, illustrating that the form of the selection window (**Equation 69**) can well describe the effects of selection in the FACS machine. Moreover, we see that the distributions $p(x)$ and $p'(x)$ are typically significantly wider than the selection window $f(x|\mu_*, \tau)$. Moreover, the fitted values of $\tau$ are almost perfectly proportional to the fraction of the population that was selected, with a value of $\tau \approx 0.03$ corresponding to the selection of 5% of the population that was used during the evolutionary runs. The fits also show that, although we in each experiment determine $\mu_*$ from

the expression of the same reference promoter, there is some variability in $\mu_*$ from one experiment to the next. From these six experiments, we find that on average $\langle \mu_* \rangle = 8.115$ and $\sigma(\mu_*) = 0.133$.

Thus, in each selection round the fitness of a promoter with mean $\mu$ and variance $\sigma^2$ is given by **Equation 71**, where $\tau = 0.03$ and $\mu_*$ fluctuates around $\langle \mu_* \rangle = 8.115$. The effective fitness experienced by this promoter is thus given by the *geometric* average of **Equation 71** with fluctuating values of $\mu_*$ and this is given by

$$f\left(\mu, \sigma | \langle \mu_* \rangle, \sigma(\mu_*), \tau\right) = \sqrt{\frac{\tau^2}{\tau^2 + \sigma^2}} \exp\left[ -\frac{(\mu - \langle \mu_* \rangle)^2 + \sigma(\mu_*)^2}{2(\tau^2 + \sigma^2)} \right]. \tag{76}$$

**Appendix 2—figure 3** shows a contour plot of this fitness function with the inferred parameters of $\langle \mu_* \rangle$, $\sigma(\mu_*)$ and $\tau$ as a function of the mean fitness $\mu$, and the excess noise level $\eta = \sigma^2 - \sigma_{min}^2(\mu)$, where $\sigma_{min}^2(\mu)$ is the minimal variance as a function of mean expression level $\mu$ (**Equation 67**). Note that, for these measurements, the plasmids were transformed into a different strain than those used to compare with the native *E. coli* promoters. We noticed that the minimal noise level $\sigma_{min}^2(\mu)$ as a function of mean $\mu$ is slightly different. Although the background fluorescence and burst size parameter $\beta = 450$ are the same, the parameter $\sigma_{ab}^2$ is smaller, that is, $\sigma_{ab}^2 = 0.006$ instead of $\sigma_{ab}^2 = 0.025$.

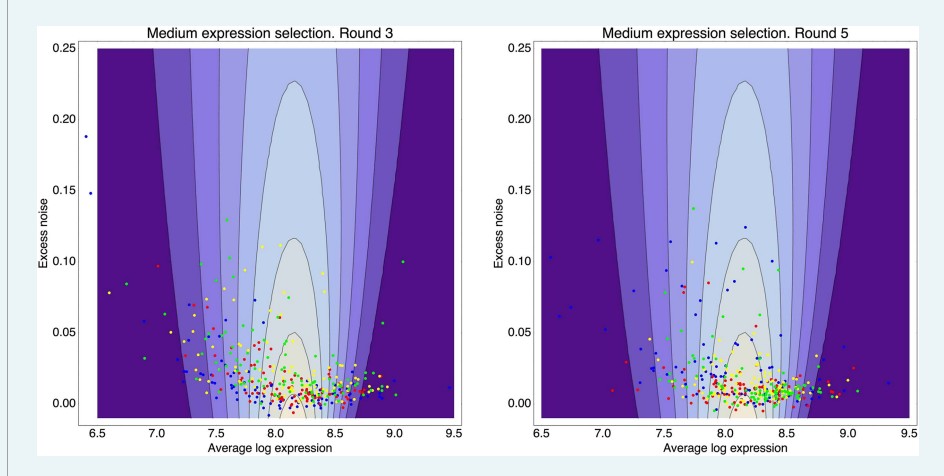

**Appendix 2—figure 3**. Contour plot of the inferred fitness function (76) as a function of mean expression $\mu$ (horizontal axis) and excess noise (vertical axis), that acts on the population from rounds 3 through 5 of the evolutionary runs for medium expression. The contours correspond to fitness values (fraction of cells selected) of 0.01, 0.02, 0.03, through 0.08. Left panel: In addition to the fitness function (contours), the panel shows the means and excess noise levels of a selection of clones from the third round of the evolutionary run (blue dots) and clones that resulted from subjecting this population to another round of selection, selecting either for the 1% (red dots), 5% (yellow dots), or 25% (green dots) of cells with expression closest to the desired expression level. Right panel: As in the left panel, but with the dots corresponding to clones from the fifth round of the evolutionary run and clones resulting from additional rounds of selection on this population (colors as in the left panel).

**Appendix 2—figure 3** clearly shows that fitness drops far more dramatically as a function of mean $\mu$ than as a function of excess noise $\eta$, that is, except for right at the optimal mean $\mu_*$, the contours are running almost vertically in the plot. Second, we do not observe promoters with high excess noise, even though their fitness would easily allow it. For example, a promoter with mean expression near the optimum 8.11 but excess noise as high as 0.25 (i.e., significantly higher than observed for any of the clones) would have higher fitness than any promoter with mean less than $\mu = 7.76$ or larger than $\mu = 8.47$ (independent of their noise), and we do observe

many promoters with means that deviate this far from the optimum. The fact that we do not observe high excess noise promoters, even though they would not be selected against, strongly suggests that such high noise promoters are uncommon a priori, that is, among all random sequences that drive expression at a medium level, the large majority have low excess noise levels and high noise promoters are rare. Moreover, note also that, for promoters that are not near the optimum, the optimal excess noise level is typically *larger* than those of the observed clones, for example, the optimal excess noise for a promoter with mean $\mu = 7.5$ is $\eta = 0.22$. These observations all indicate that promoters have not experienced significant selection on their noise levels.

To further support this conclusion, **Appendix 2—figure 4** shows the inferred fitness values for the observed clones both as a function of their mean expression (left panel) and as a function of their excess noise (right panel). The figure shows that, whereas the fitness of a clone can be accurately predicted from its mean $\mu$, fitness is almost entirely uncorrelated to a promoter's excess noise $\eta$.

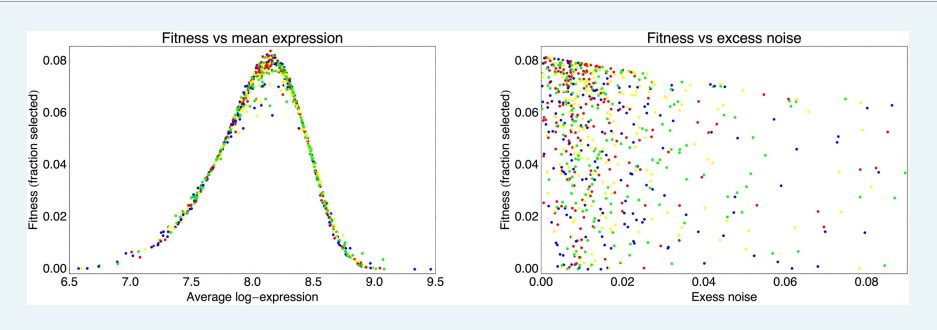

**Appendix 2—figure 4**. Fitness of the observed clones, as given by **Equation 76**, as a function of their mean expression $\mu$ (left panel) and their excess noise $\eta$ (right panel). As in **Appendix 2—figure 3**, the blue dots correspond to clones from the third and fifth round of the evolutionary run, the red dots result from another round of stringent selection (top 1%), the yellow dots from another round of standard selection (top 5%), and the green dots from a round of weaker selection (top 25%).

Finally, if there was significant selection on noise levels, then we expect noise levels to systematically shift under selection. **Appendix 2—figure 5** shows cumulative distributions of excess noise levels for clones obtained from different populations of cells.

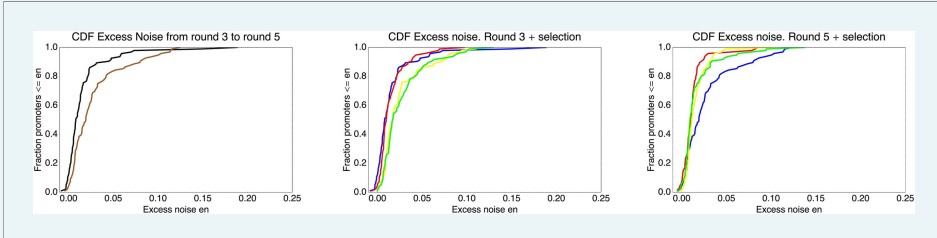

**Appendix 2—figure 5**. Cumulative distribution functions of excess noise levels for the promoters extracted from different populations. Left panel: Excess noise levels of promoters from the third (black) and fifth (brown) round of the evolutionary run. Middle panel: Excess noise levels of promoters from the third round of the evolutionary run (blue), and from clones that resulted from another round of either stringent (red), normal (yellow), or weak (green) selection. Right panel: As in the middle panel but now for clones from the fifth round and clones resulting from another round of selection on this population.

The figure shows that, surprisingly, the excess noise levels seem to increase from the third to the fifth round in the evolutionary run. However, given the limited number of clones involved,

the change in excess noise levels is only marginally significant ($p = 0.004$ in a t-test). Similarly, the effect of selection on excess noise levels seems to be opposite on the populations of round 3 and round 5 (center and right panels in *Appendix 2—figure 5*). We suspect that there are some systematic experimental fluctuations that make measured excess noise levels vary across days, and that the observed distributions of excess noise levels are more a reflection of experimental variability than of true shifts in the distribution. Importantly, excess noise levels larger than 0.1, which are observed for a substantial fraction of native promoters, are very rare for all these populations.

In summary, our in-depth analysis of the FACS fitness function and the effects of selection show that the noise properties of the synthetic promoters have not been significantly shaped by selection. Already the synthetic promoters at the third round of selection are tightly concentrated in a low noise band, even though selection does not select against noise, and promoters with mean away from the optimum would benefit from having higher noise. Two additional rounds of mutation and selection on the promoters from the third round do not substantially change the distribution of noise levels, confirming that there are no substantial fitness differences among promoters with different noise. Similarly, performing additional rounds of selection (be it very stringent, normal, or lenient) also does not substantially change the observed noise levels of the selected promoters. Thus, our results show that promoters selected from a large collection of random sequences naturally display low noise levels. Importantly, this implies that the native promoters with substantially higher noise levels must have experienced some selective pressures that caused them to increase their noise.

**Appendix 3**

# A simple model for the evolution of gene regulation and expression noise

Given a particular environment, the fitness, for example, growth rate or survival probability of a cell, depends on the expression level of its genes. Note that the fact that gene regulatory mechanisms have evolved already demonstrates that different environments require different gene expression patterns, that is, expressing a gene at the 'wrong' level for a given environment has negative effects on fitness/growth rate. Although our model can be developed for an arbitrary number of genes, for simplicity we will focus on a single gene. We assume that, in each given environment, there is an optimal expression level $\mu_\star$. Given that, as we have seen, expression levels are roughly log-normally distributed, we will express expression levels in log-space, that is, the logarithm of the number of proteins per cell. We define the fitness at the optimal expression level $\mu_\star$ as $f_o$. Fitness will fall as the expression level $x$ moves away from this optimum. In this simple conceptual model, we will assume that, like in our FACS selection, the fitness $f(x)$ falls approximately Gaussian away from the optimum, that is,

$$f(x|\mu_\star, \tau) = f_o \exp\left[-\frac{1}{2}\left(\frac{x-\mu_\star}{\tau}\right)^2\right],$$

(77)

where $\tau$ is again a parameter that determines how fast the fitness falls when the expression $x$ moves away from the optimum $\mu_\star$.

To justify the Gaussian form of the fitness function, assume that expression levels are maintained for some characteristic time $t$. Over this time, cells that grow at rate $\rho$ will expand by a factor of $f = e^{\rho t}$. The growth rate $\rho$ is optimal at $x = \mu_\star$, and to second order in the difference between $x$ and $\mu_\star$, we can write

$$\rho = \rho_o - \frac{1}{2}(x-\mu_\star)^2\left\|\frac{d^2\rho}{dx^2}\right\|_{x=\mu_\star}.$$

(78)

Defining $f_o = e^{\rho t}$ and $1/\tau^2 = t\left\|\frac{d^2\rho}{dx^2}\right\|_{x=\mu_\star}$, we obtain the fitness function defined above. Note that, in this interpretation, the width of the selection function is inversely proportional to the time for which fitness fluctuations are maintained.

In complete analogy with the FACS selection case, the fitness $f(\mu, \sigma|\mu_\star, \tau)$ of a promoter with mean $\mu$ and variance $\sigma^2$ in an environment with optimal expression level $\mu_\star$ and width $\tau$ is given by the integral of the product of the fitness function (**Equation 77**) and the Gaussian distribution of expression levels, giving

$$f(\mu, \sigma|\mu_\star, \tau) = \sqrt{\frac{\tau^2}{\tau^2+\sigma^2}}\exp\left[-\frac{1}{2}\frac{(\mu-\mu_\star)^2}{\tau^2+\sigma^2}\right].$$

(79)

Note that this functional form is a good approximation to the fitness function as long as expression levels are roughly log-normally distributed and as long as the integral of expression levels and fitness function can be approximated using the standard Laplace approximation, that is, expanding the logarithm of fitness to second order around its maximum.

We now extend this simple situation in two respects. First, instead of assuming that the optimal level $\mu_\star$ is fixed, we imagine that the population of cells has gone through several different 'environments' where, in each environment $e$, there was an optimal expression level $\mu_e$. Although we could develop our theory assuming the width $\tau$ varies across environments, the

analytical results are much more transparent if we assume $\tau$ is the same in each environment, and we will make this assumption for clarity of presentation.

Let us first consider what this situation implies for the fitness of a promoter expressing at constant mean level $\mu$ with variance $\sigma^2$. The number of offspring that a strain with mean $\mu$ and variance $\sigma^2$ produces (or leaves behind) after experiencing environment $e$ is proportional to $f(\mu, \sigma | \mu_e, \tau)$. Consequently, the final number of offspring produced after experiencing all environments is given by the product $\prod_e f(\mu, \sigma | \mu_e, \tau)$. We define the overall log-fitness log[$f(\mu, \sigma)$] as the average of the log-fitness across all environments:

$$\log[f(\mu, \sigma)] = \langle \log[f(\mu, \sigma | \mu_e, \tau)] \rangle_e, \tag{80}$$

where the subscript $e$ indicates that we are averaging over all environments $e$ (which we drop for convenience from here on). Using **Equation 79**, we obtain

$$\log[f(\mu, \sigma)] = -\frac{\text{var}(\mu_e) + (\langle \mu_e \rangle - \mu)^2}{2(\sigma^2 + \tau^2)} + \frac{1}{2}\log\left[\frac{\tau^2}{\sigma^2 + \tau^2}\right], \tag{81}$$

where $\langle \mu_e \rangle$ is the average and $\text{var}(\mu_e)$ is the variance in desired expression levels across the environments. It is immediately clear from **Equation 81** that, as a function of the mean expression $\mu$, optimal fitness is obtained when $\mu = \langle \mu_e \rangle$. Substituting this optimal mean level, we find that the optimal variance is given by

$$\sigma^2 = \text{var}(\mu_e) - \tau^2, \tag{82}$$

when $\text{var}(\mu_e) \geq \tau^2$, and $\sigma = 0$ otherwise; that is, when the variance in desired expression levels is larger than the width of the selection window $\tau$, then the fitness of a constantly expressed promoter can be increased by raising the noise level $\sigma$ of the promoter until the sum $\sigma^2 + \tau^2$ equals the variation in desired levels $\text{var}(\mu_e)$. This result is equivalent to results on selection for phenotypic variance obtained previously (for example, **Bull, 1987**; **Haccou and Iwasa, 1995**). However, in these previous models that more abstractly considered 'phenotypic traits', it was assumed that both the mean and variance of the phenotypic trait were not only directly encoded by the genotype but could also be independently altered through mutations, without explicitly considering how mean and variance would be encoded in the genotype. In our case, where the 'trait' under study is the transcription rate of a promoter, it is a priori quite clear how mutations may alter mean levels, for example, through changes in the affinity of the sigma-factor binding site, but much less clear how the variance is encoded in the genotype. Moreover, rather than simply increasing its noise, we would naturally expect that promoters would evolve gene regulation in order to deal with different required expression levels across different environments.

## Including gene regulation

We now further extend the model by considering that there are various transcriptional regulators in the cell whose activities may vary across the different environments $e$. By evolving binding sites for a TF, the promoter becomes regulated by it and, consequently, the mean expression $\mu$ becomes a function $\mu(e)$ of the environment $e$.

For simplicity we first consider the case of a single regulator whose mean activity (i.e., concentration of the DNA-binding version of the regulator) $r_e$ is a function of the environment $e$. Since the transcriptional regulator's expression will itself also be subject to gene expression noise, the activity of the regulator varies from cell to cell. We will assume that, in each environment, the activity of the regulator varies from cell to cell in a roughly Gaussian manner with variance $\sigma_r^2$, that is, the probability to find a cell in environment $e$ with regulator level $r$ is

$$P(r | r_e, \sigma_r^2) = \frac{1}{\sqrt{2\pi}\sigma_r} \exp\left[-\frac{(r - r_e)^2}{2\sigma_r^2}\right]. \tag{83}$$

We characterize the regulation of the promoter by the regulator through a single coupling constant $c$ such that, in cells with regulator level $r$, the distribution of expression levels is a Gaussian with mean $\mu + cr$ and variance $\sigma^2$, that is,

$$P(x|\mu,\sigma^2,r) = \frac{1}{\sqrt{2\pi}\sigma}\exp\left[-\frac{(x-\mu-cr)^2}{2\sigma^2}\right]. \tag{84}$$

Thus we assume the expression level of the target is a linear function of the activity of the regulator. Integrating over the distribution of regulator levels $P(r|r_e,\sigma_r^2)$, the final distribution of expression levels is given by a Gaussian with mean $\mu(e) = \mu + cr_e$ and variance $\sigma^2 + c^2\sigma_r^2$, that is,

$$P(x|\mu,\sigma^2,c) = \frac{1}{\sqrt{2\pi(\sigma^2+c^2\sigma_r^2)}}\exp\left[-\frac{(x-\mu-cr_e)^2}{2(\sigma^2+c^2\sigma_r^2)}\right]. \tag{85}$$

In environment $e$, with desired level $\mu_e$, the fitness of a promoter with coupling $c$ is then given by

$$f(\mu,\sigma,c|\mu_e,\tau) = \sqrt{\frac{\tau^2}{\sigma^2+c^2\sigma_r^2+\tau^2}}\exp\left[-\frac{(\mu+cr_e-\mu_e)^2}{2(\sigma^2+c^2\sigma_r^2+\tau^2)}\right]. \tag{86}$$

Finally, the log-fitness, averaged over all environments $e$, is given by

$$\log[f(\mu,\sigma,c)] = -\frac{1}{2}\frac{\langle(\mu+cr_e-\mu_e)^2\rangle}{\tau^2+\sigma^2+c^2\sigma_r^2} + \frac{1}{2}\log\left[\frac{\tau^2}{\tau^2+\sigma^2+c^2\sigma_r^2}\right]. \tag{87}$$

It is again easy to see that, with respect to the mean expression $\mu$, fitness is optimized when $\mu = \langle\mu_e\rangle - c\langle r_e\rangle$. In the following we will assume that the mean expression $\mu$ matches this optimal value.

We can rewrite the expression for the average log-fitness in a simpler form by introducing the following set of effective parameters. First, the variable

$$Y^2 = \frac{\mathrm{var}(\mu_e)}{(\tau^2+\sigma^2)}, \tag{88}$$

measures the variance in desired expression levels $\mu_e$ relative to the sum of the variances associated with the width of selection $\tau^2$ and the noise of the unregulated promoter $\sigma^2$. The variable $Y$ quantifies the 'expression mismatch' between the promoter's average expression $\mu$ and the (varying) desired expression levels $\mu_e$. Notably, in the absence of coupling to a regulator, the log-fitness is $-Y^2/2 + \log[\tau^2/(\sigma^2 + \tau^2)]/2$, that is, the higher $Y^2$, the lower the fitness of the unregulated promoter.

The second effective parameter

$$X^2 = \frac{c^2\sigma_r^2}{(\tau^2+\sigma^2)}, \tag{89}$$

measures the strength of the regulator coupling constant $c$. More precisely, it quantifies the contribution $c^2\sigma_r^2$ to the promoter's variance in gene expression, again relative to $(\sigma^2 + \tau^2)$, that is, it measures the strength of the noise propagation. We will refer to $X$ as the coupling constant. Third, the parameter

$$S^2 = \frac{\mathrm{var}(r_e)}{\sigma_r^2}, \tag{90}$$

measures the 'signal-to-noise' ratio of the regulator, that is, the variance $\text{var}(r_e)$ of its mean level across conditions relative to its variance $\sigma_r^2$ within each condition. A regulator with large $S$ varies a lot in activity across environments and has relatively little noise in each, whereas a regulator with small $S$ varies little across environments relative to its noise level. Finally, we have the correlation $R$ between the desired expression levels $\mu_e$ and the regulator's activities $r_e$, that is,

$$R = \frac{\langle \mu_e r_e \rangle - \langle \mu_e \rangle \langle r_e \rangle}{\sqrt{\text{var}(r_e)\text{var}(\mu_e)}}. \tag{91}$$

In terms of these parameters, we have for the average log-fitness

$$\log[f(X, Y, S, R)] = -\frac{1}{2}\frac{Y^2(1 - R^2) + (SX - RY)^2}{1 + X^2} - \frac{1}{2}\log[1 + X^2] + \frac{1}{2}\log\left[\frac{\tau^2}{\tau^2 + \sigma^2}\right]. \tag{92}$$

The last term is a constant that does not depend on our effective parameters and we will ignore it from now on.

We intuitively expect that the promoter's fitness would benefit most from coupling to a regulator that is perfectly correlated with the environment's requirements, that is, at $R = 1$. Indeed, we find that the derivative $\log[f(X)]$ with respect to $R$ is given by

$$\frac{d\log[f(X, Y, S, R)]}{dR} = \frac{SXY}{1 + X^2}, \tag{93}$$

which is positive as long as the desired levels vary ($Y > 0$), the regulator has some variation across environments ($S > 0$), and there is positive coupling ($X > 0$). Thus, in general, if we keep all other variables fixed, an increase in the regulator's correlation $R$ is always beneficial.

We now consider the case in which a regulator with a given correlation $R$ and signal-to-noise rate $S$ is given, and we want to determine the optimal coupling $X_*$ that maximizes $\log[f(X, Y, S, R)]$. The derivative of $\log[f(X, Y, S, R)]$ with respect to $X$ is given by

$$\frac{d\log[f(X, Y, S, R)]}{dX} = \frac{XY^2 - X(1 + X^2 + S^2) + SR(1 - X^2)}{(1 + X^2)^2}. \tag{94}$$

At $X = 0$, this derivative equals $SR$. Thus, whenever $R > 0$, the derivative is positive at $X = 0$. Because, as can be easily seen from **Equation 94**, the derivative is guaranteed to be negative for large $X$, this implies that, whenever $R > 0$, there is an optimal coupling $X_*$ that is positive, that is, $X_* > 0$. Thus, whenever $R > 0$, the promoter is guaranteed to increase its fitness by evolving a non-zero coupling to the regulator.

The optimal coupling $X_*$ is given by the positive solution of the third order polynomial in the numerator of **Equation 94**. In general we find that, when $Y$ is small, the optimal coupling is given by

$$X_* = \alpha_0 Y = \frac{SR}{1 + S^2}Y, \tag{95}$$

and that, when $Y$ is very large, $X_*$ obeys

$$X_* = \alpha_\infty Y = \left(\sqrt{\left(\frac{SR}{2}\right)^2 + 1} - \frac{SR}{2}\right)Y. \tag{96}$$

That is, both for very small and very large $Y$, the optimal coupling $X_*$ is directly proportional to $Y$, with proportionality constants $\alpha_0$ and $\alpha_\infty$, respectively. Moreover,

$\alpha_\infty \geq \alpha_0$. The behavior of $X_*$ as a function of $Y$ for different values of $R$ and $S$ is illustrated in **Appendix 3—figure 1**.

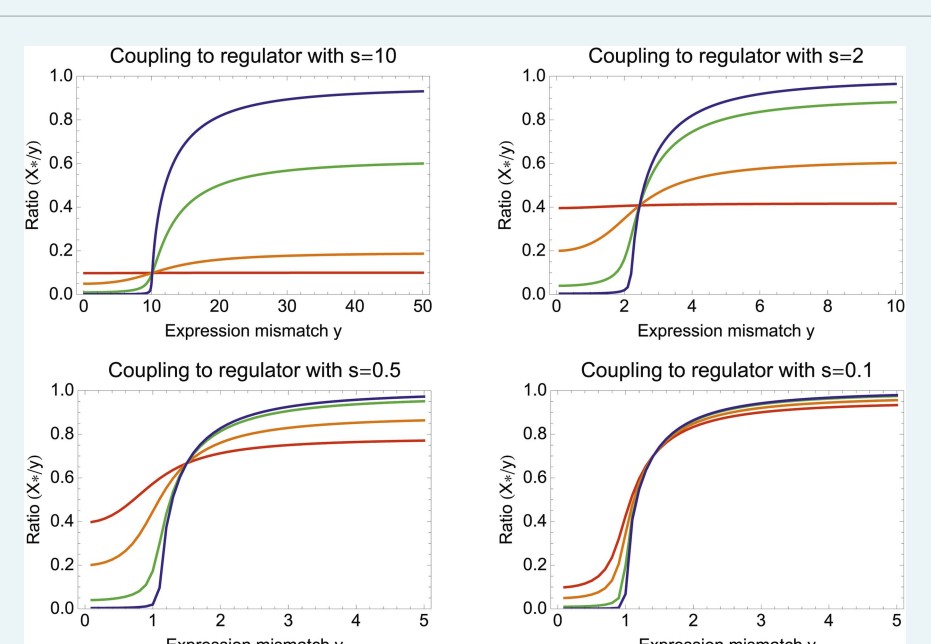

**Appendix 3—figure 1**. The ratio $X_*/Y$ between optimal coupling $X_*$ and expression mismatch $Y$ as a function of $Y$, for different values of the regulator's signal-to-noise ratio $S$ and the correlation between regulator and environment $R$. Each panel corresponds to a different signal-to-noise ratio $S$, from a high signal regulator in the top left to a noisy regulator at the bottom right. In each panel, the different colored lines correspond to different correlations $R$, that is, $R = 0.01$ (blue), $R = 0.1$ (green), $R = 0.5$ (orange), and $R = 0.99$ (red).

The figure shows that, as $Y$ increases, the ratio $X_*/Y$ switches from the lower $\alpha_0$ to the higher $\alpha_\infty$. Whenever both the correlation $R$ and the signal-to-noise $S$ are high (orange and red curves in the top two panels), there is only a small difference between $\alpha_0$ and $\alpha_\infty$, that is, $X_*$ increases roughly linearly with $Y$ when there is a well-correlated regulator with high signal-to-noise.

In contrast, when the correlation $R$ is low or the regulator is noisy, there is a large difference between $\alpha_\infty$ and $\alpha_0$. Moreover, the optimal coupling shows a sharp transition from low values to much higher values at a 'critical' value of $Y$. This critical value of $Y$ occurs at $Y = \sqrt{1 + S^2}$ when $R$ is low (blue and green curves), and slightly earlier when $R$ increases (orange and red curves). When the regulator is very noisy (bottom right panel of **Appendix 3—figure 1**) the behavior of $X_*$ becomes almost independent of the correlation $R$, showing a sharp transition from almost no coupling to strong coupling when $Y \approx 1$. This behavior even extends to the case where there is no correlation whatsoever between the regulator and the environment, that is, $R = 0$.

## Coupling to an uncorrelated regulator
When $R = 0$ the optimal coupling is given by

$$X_*^2 = \max\left[0, Y^2 - 1 - S^2\right]. \tag{97}$$

That is, at the critical value $Y = \sqrt{1 + S^2}$ the coupling goes from zero to a positive value. For large $Y$, the optimal coupling is simply $Y$. The behavior of optimal coupling $X_*$ as a function of $Y$ is shown in **Appendix 3—figure 2**.

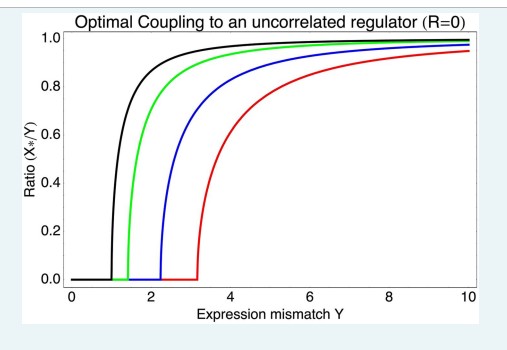

**Appendix 3—figure 2**. Optimal coupling $X_*$ to an uncorrelated regulator ($R = 0$) as a function of the expression mismatch $Y$ for different values of the signal-to-noise ratio $S$, that is, $S = 0$ (black), $S = 1$ (green), $S = 2$ (blue), and $S = 3$ (red).

## Log-fitness at optimal coupling $X_*$

We next consider the case in which the promoter has a certain expression mismatch $Y$, and we calculate the log-fitness that it can obtain by optimally coupling to a regulator that has a certain signal-to-noise $S$ and correlation $R$. **Appendix 3—figure 3** shows the resulting log-fitness values as a function of $S$ and $R$ for four different values of the expression mismatch $Y$: $Y = 1$ in the top left panel, $Y = 2$ in the top right panel, $Y = 4$ in the bottom left panel, and $Y = 8$ in the bottom right panel.

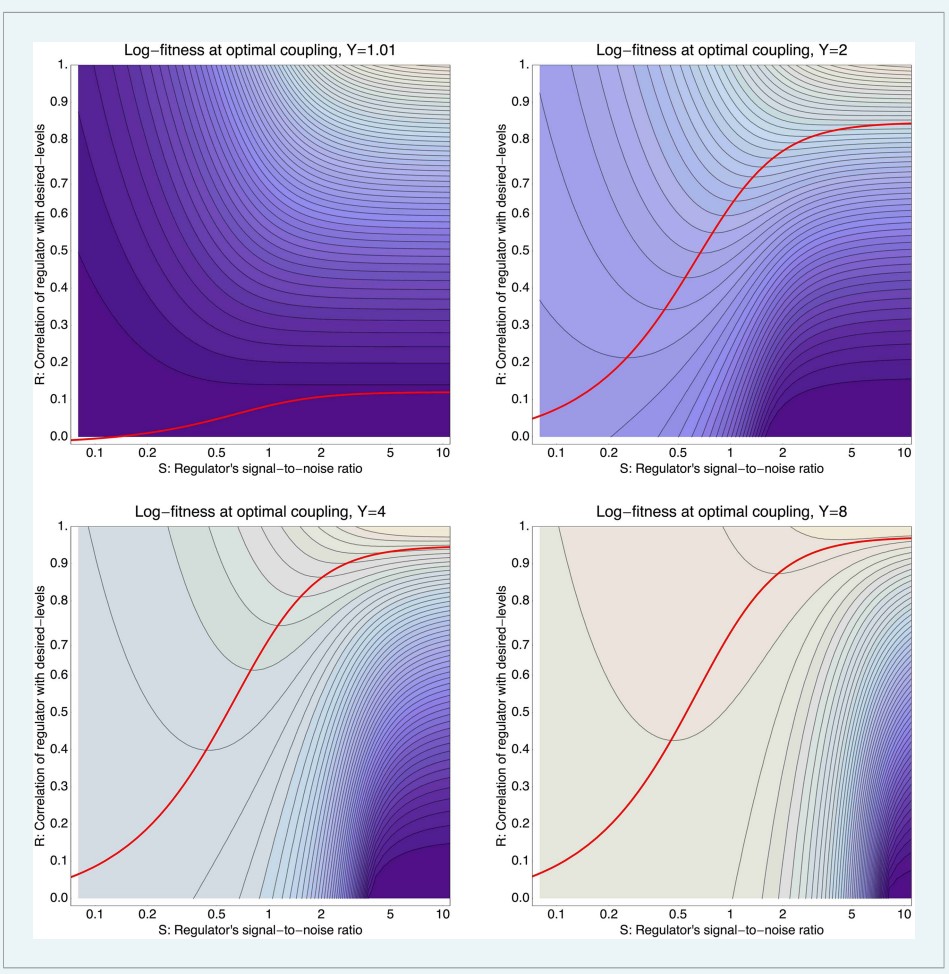

**Appendix 3—figure 3**. Log-fitness as a function of the signal-to-noise ratio $S$ (horizontal axis) and correlation $R$ of the regulator (vertical axis) for a promoter that is optimally coupled ($X = X_*$) to the regulator. The different panels correspond to log-fitnesses that are obtained for different values of the expression mismatch $Y$ (indicated in the title of each panel). The contours run from $-0.04$ to $-0.5$ in the top left panel, from $-0.3$ to $-1.9$ in the top right panel, from $-1$ to $-12$ in the bottom left panel, and from $-2$ to $-30$ in the bottom right panel. The red curves show optimal signal-to-noise $S$ as a function of the correlation $R$.

When the expression mismatch is small, that is, $Y = 1$ corresponding to a variance var($\mu_e$) that matches $\sigma^2 + \tau^2$, then fitness generally increases with increasing $R$ and $S$. However, the absolute value of the fitness increase is small, that is, for small $Y$, promoters already have reasonably high fitness without regulation. As the value of $Y$ increases, the log-fitness values start varying more dramatically as a function of $R$ and $S$. Independent of the value of $Y$, the optimal fitness is always obtained for very high $R$ and high $S$. In this regime the regulators are very accurate (high $S$) and correlate very well with the desired expression levels of the promoters (high $R$). In this regime the noise-propagation is very small and the condition-response effect of the regulator is high, that is, as expected, the most optimal solution is a very accurate regulation in which all benefits of the regulation stem from the condition-response effect. However, when $Y$ is large, an almost equally high fitness can be obtained by coupling to a 'noisy' regulator with low $S$ and $R = 0$ in the bottom left of the plots. In this regime the condition-response effect is small and noise-propagation is large, and the benefits of the regulatory interaction stem solely from the noise-propagation, that is, by coupling to the noisy regulator a bet hedging strategy is implemented. Notably, when $Y$ is large, only regulators with very high correlation $R$ and large signal-to-noise $S$ can outcompete coupling to an entirely noisy regulator with $R = 0$ and small $S$; that is, to outcompete noise-propagation, the condition-response effect has to be highly accurate. In other words, unless a regulator is available that very precisely regulates the promoter to attain its desired expression levels, best fitness can often be obtained through noise-propagation.

Note also that, whenever $Y$ is larger than 1 and the correlation $R$ is not very high, fitness generally decreases rapidly with the signal-to-noise of the regulator; that is, when a regulator has only moderate correlation with the desired expression levels of its target, low signal-to-noise is preferred. This suggests that regulators that are regulating targets whose desired expression levels correlate only moderately with the regulator's activity may be under selection for *lowering* their signal-to-noise ratios.

To illustrate the selection on the signal-to-noise level of the regulator, the red curves in **Appendix 3—figure 3** show the optimal signal-to-noise $S_*$ as a function of the correlation $R$. To the left of these lines the signal-to-noise is too low, that is, the regulator is too noisy, and fitness can be increased making the regulator more accurate. In this regime the noise-propagation is a detrimental side-effect of the regulatory interaction. For each value of $Y$ there is a critical value of $R$ where the optimal signal-to-noise $S_*$ diverges. Above this critical value of $R$ the noise-propagation is always detrimental. However, unless $Y$ is close to 1, this critical value of $R$ is generally very high.

To the right of the red curves in **Appendix 3—figure 3** the accuracy of the regulator is too high for its correlation $R$, and fitness could be improved by increasing the noise of the regulator. In this regime the noise-propagation is not large enough. Interestingly, the red curve thus corresponds to a continuum of regulatory strategies where the condition-response and noise-propagation are optimally acting in concert. This suggests a plausible scenario for evolving accurate regulation de novo from a state without regulation. Initially, a promoter can couple to a noisy regulator whose activities do not correlate at all with the desired expression levels of the promoters. Due to the noise-propagation, this coupling may already be beneficial. Once the regulatory interaction exists, the coupling strength, the activity profile of the regulator, and its noise level can be slowly mutated so as to increase both the correlation $R$ and signal-to-noise $S$ along the red curve, leading finally to accurate regulation where the condition-response effect dominates.

These considerations also suggest different possible scenarios for the joint evolution of multiple promoters and their regulators. On the one hand, when a regulator is coupled to a single promoter or a set of promoters whose desired expression levels are perfectly correlated across environments, it can increase overall fitness by increasing the correlation $R$ between the regulator's activities and the desired expression levels of its targets. In this way, regulation may evolve to become more precise over time. On the other hand, promoters may often have an incentive to couple to regulators that only moderately correlate with their desired levels. Once a regulator is coupled to multiple promoters that have different desired expression levels, there is no way that the regulator can adapt its activities to correlate highly with the desired levels of all its targets, and such regulators will experience selection to become more noisy.

## Final noise levels under optimal coupling and signal-to-noise

We next consider what final noise levels $\sigma_{\text{tot}}^2 = \sigma^2 + c^2\sigma_r^2$ result when a promoter, with a certain expression mismatch $Y$, couples optimally to a regulator which has a certain correlation $R$, and whose signal-to-noise level has been optimized as well.

To this end we need to determine the jointly optimal coupling $X_*$ and signal-to-noise $S_*$ given a certain expression mismatch $Y$ and correlation $R$. From **Equation 92** it is easy to see that fitness is maximized with respect to the signal-to-noise level $S$ when

$$S_* = \frac{RY}{X}. \tag{98}$$

If we substitute this value back into **Equation 92**, we find that the optimal coupling $X_*$ is now given by

$$X_*^2 = \max\left[0, \left(1 - R^2\right)Y^2 - 1\right]. \tag{99}$$

Note that $Y^2$ is the variation in desired expression levels and $R^2Y^2$ is the amount of this variation that the condition-response effect manages to 'track' when the promoter is coupled to the regulator. Thus, $(1 - R^2)Y^2$ is precisely the remaining variance in desired expression levels that the condition-response is unable to track. To bring this out more clearly, we substitute back our original parameters. We then find that when $(1 - R^2)Y^2 < 1$:

$$\sigma_{\text{tot}}^2 = \sigma^2, \tag{100}$$

and when $(1 - R^2)Y^2 > 1$:

$$\sigma_{\text{tot}}^2 = \left(1 - R^2\right)\text{var}(\mu_e) - \tau^2. \tag{101}$$

This brings out most clearly that, when the regulation is imprecise ($(1 - R^2)Y^2 > 1$), the final noise level that is evolved matches the fraction of the variance in desired expression levels $\text{var}(\mu_e)$ that is not tracked by the condition-response. In other words, the evolved transcriptional noise level of a promoter precisely reflects to what extent the promoter's regulation is not able to track the expression levels desired by the environment.

*Figure 3—figure supplement 1* shows the total noise level $\sigma_{\text{tot}}$ as a function of $Y$ and $R$ when the promoter is optimally coupled to a regulator with optimal signal-to-noise. The figure illustrates that there are two regimes of solutions ('phases') separated by a phase boundary (thick black curve) that occurs at $(1 - R^2)Y^2 = 1$. On one side of this boundary, in the top left of the figure, the final noise level $\sigma_{\text{tot}}$ is essentially not different from the original noise level $\sigma$. This occurs either when $Y < 1$, that is, when no regulation is necessary, or when very accurate regulation is available. Note that, similarly to what we saw in the last section, very high correlations $R$ are necessary to realize this regime at larger values of $Y$. We call this the 'basal noise regime'.

The largest part of the parameter space occurs on the other side of the phase boundary, which we call the 'environment-driven noise regime'. Here the final noise level $\sigma_{tot}$ becomes independent of the original noise $\sigma$, but is instead determined by the fraction of variation in desired expression levels var($\mu_e$) that is not tracked by the condition-response, that is, by $(1 - R^2)$ var($\mu_e$). The figure also indicates the optimal values of $S_*$ as a function of $Y$ and $R$. The optimal signal-to-noise $S_*$ diverges at the phase boundary, that is, in the basal noise regime, regulators are preferred with signal-to-noise that is as high as possible. In contrast, for the majority of the parameter space in the environment-driven noise regime, signal-to-noise levels of 1 or less are preferred, that is, unless regulation is very precise, noisy regulators are typically preferred over precise regulators.

The figure also demonstrates that, unless $R$ is close to 1, the final noise increases with the variance in desired expression levels var($\mu_e$). Thus, unless there is a systematic correlation between the expression mismatch $Y$ of a promoter and the correlation of the regulator with highest available correlation, noise levels are expected to increase with the 'plasticity' var($\mu_e$) that the environment requires of the promoters.

Similarly, the larger $Y$, the larger the remaining variance $Y'^2 = (1 - R^2)Y^2$ tends to be after coupling to the regulator with the highest available correlation $R$. Whenever this remaining expression mismatch is $Y'$ large, the promoter will have an incentive to couple to further regulators, that is, the theory also generally predicts that promoters with high var($\mu_e$) tend to couple to more regulators.

Finally, we note that this theoretical model can easily be extended to the case of multiple promoters and regulators. In particular, because in our model the promoter's expression is a linear function of the regulatory inputs, the theory extends easily to promoters coupling to multiple regulators with different coupling constants. However, the regulatory network structure that will evolve in this general case will depend crucially on the correlation structure of the desired expression levels across all the promoters. Moreover, there might be many environmental changes that affect the optimal expression levels but that cannot be sensed by any of the regulators, and this will constrain the extent to which regulators can optimize their activities to match the desired levels of their targets. We intend to pursue such analyses in future work.

