## [Decision Letter]

Thank you for sending your work entitled “Expression noise facilitates the evolution of gene regulation” for consideration at *eLife*. Your article has been favorably evaluated by Aviv Regev (Senior editor), a guest Reviewing editor, and two reviewers.

The Reviewing editor and the reviewers discussed their comments before we reached this decision, and the Reviewing editor has assembled the following comments to help you prepare a revised submission:

We all thought that the subject matter of the work was very timely, and the findings interesting. There were, however, a number of significant flaws that need to be addressed before publication is considered.

1) After more than a decade of intensive research, there has been significant progress towards a biophysical, molecular understanding of the mechanisms leading to transcription noise. See e.g. the review by Sanchez and Golding (Science 2013), and more recent work in *E. coli* from the Xie (Cell 2014) and Phillips (Science 2014) labs. In its current form, the authors' work appears largely disconnected from our current mechanistic understanding of transcription noise: How does this understanding motivate the work? How does it affect the interpretation of the experimental results? How does it motivate the theoretical model and its interpretation?

2) A possibly related flaw is the rather weak connection between the experiments and theoretical model. First, the model itself should be described in some more detail in the main text. But more crucially: How does the model help interpret the experiments? Can any kind of quantitative comparison be made between the two sets of results?

3) More care must be taken when defining and discussing “transcription noise”. In the current work, neither transcription kinetics (e.g. using MS2-GFP) nor mRNA statistics (using single-molecule FISH) are directly measured, but instead a protein (GFP) reporter is used. Thus, interpreting the measured variability as “transcription noise” involves important caveats (see e.g. Paulsson, Nature 2004; Pedraza and Paulsson, Science 2008). These need to be explicitly discussed.

4) In multiple places, the authors make very strong statements, which require additional evidence to support them or otherwise need to be reworded. Examples:

“Many studies have used *circumstantial evidence* to suggest that selection generally acts to minimize noise […] there is *empirical evidence* that selection has acted to increase expression noise in some cases” (Introduction). What makes the former studies merely “circumstantial” while the latter rise to the level of “empirical”?

“Whenever fluctuations are small relative to the mean, which applies to most promoters” (Results). No reference is given to support this statement.

“We concluded that constitutively expressed *E. coli* promoters exhibit low excess noise levels by default” (Results). How was this conclusion reached? In particular, what is a “constitutive promoter” (or “unregulated promoter”, as mentioned later in the same section of the manuscript)? All known promoters in *E. coli* are regulated by multiple parameters: growth rate, nutritional state, temperature, etc. So how can a specific promoter be “constitutive”/“unregulated”?

---

## [Author Response]

*1) After more than a decade of intensive research, there has been significant progress towards a biophysical, molecular understanding of the mechanisms leading to transcription noise. See e.g. the review by Sanchez and Golding (Science 2013), and more recent work in* E. coli *from the Xie (Cell 2014) and Phillips (Science 2014) labs. In its current form, the authors' work appears largely disconnected from our current mechanistic understanding of transcription noise: How does this understanding motivate the work? How does it affect the interpretation of the experimental results? How does it motivate the theoretical model and its interpretation*?

We agree with the reviewers that, for the sake of succinctness, we left out a discussion of the current understanding of mechanisms causing expression noise. To better put our work into context, we have now included such a discussion in the Introduction and cited some of the works the reviewers mention.

As the reviewers point out, there is by now a large literature on the molecular mechanisms involved in gene expression noise, and since we obviously cannot give a comprehensive overview of this literature, we focused on the points that are most relevant for our work. Briefly, from a mechanistic perspective, it is well accepted that, because a given promoter is only present in one or a few copies within a single cell, and because all molecules in the cell are subject to Brownian motion and thermal fluctuations, the sequence of transcription initiation events from a given promoter is inherently stochastic. In the simplest scenario both transcription initiation and mRNA decay occur at a constant probability per unit time, leading to a Poissonian distribution of transcript numbers, with variance equal to mean. However, many promoters have been shown to exhibit variance in expression that is larger than their mean, which is generally quantified using the Fano factor (ratio of variance to mean). Moreover, this Fano factor has been shown to vary significantly across different promoters. Mechanistically, this additional noise is generally understood to arise from promoters stochastically switching between different ‘states'. In the simplest models, there is only an ‘on’ and an ‘off’ state, with promoters producing bursts of transcripts in the ‘on’ state only (indeed some authors simply identify the Fano factor with ‘burst size’). However, Fano factors larger than one result whenever promoters switch stochastically between different states that have different transcription rates associated with them. Such general state changes are likely associated with events such as the binding and unbinding of different transcription factors, local changes to the state of the chromatin, and so on. Moreover, since the concentrations of TFs may vary from cell to cell, the rates of switching between different states may vary from cell to cell as well. Thus, these mechanistic considerations suggest that the level of transcriptional noise may, at least to some extent, be encoded in the sequence of the promoter. Indeed, several recent works have established this to be the case. Consequently, transcriptional noise is thus subject to evolution through natural selection, and the main question that we set out to investigate in this work is how selection has shaped noise properties of promoters.

In summary, the current understanding of the mechanisms underlying transcriptional noise are very much in line with our interpretation of the results presented in the paper: different promoter sequences may lead to different levels of transcriptional noise because they contain different sets of binding sites for regulatory proteins. Indeed as we were performing this work, several works appeared that provide further support that regulatory sites in promoters are important contributors to transcription noise (e.g. [16]; [17]; [32]), which we now also cite in the text.

*2) A possibly related flaw is the rather weak connection between the experiments and theoretical model. First, the model itself should be described in some more detail in the main text. But more crucially: How does the model help interpret the experiments? Can any kind of quantitative comparison be made between the two sets of results*?

We agree with the reviewers that the presentation of the theoretical model was rather terse and we have in the revision substantially expanded on the details of the theoretical model in the main text, including also the main analytical result illustrated in Figure 4. We also expanded the description of the model in the Materials and methods section.

Most importantly, we have altered the presentation to better explain how precisely our model helps interpret the results of our experiments. The general structure of our presentation is to first summarize the main experimental observations that we want to explain: Natural selection must have acted to increase the noise levels of a substantial fraction of *E. coli* promoters, and the promoters with elevated noise are generally characterized by being highly regulated, i.e. having known regulatory sites and showing higher expression plasticity across conditions. We note that this general association between high transcriptional noise and gene regulation has now been observed in several studies.

After this, we review the explanations that have been proposed to account for selection for noise, on the one hand, and for the general association between noise and regulation, on the other hand. We review theoretical work arguing that increased phenotypic variability can act as a beneficial ‘bet hedging’ strategy, but point out that this interpretation fails to explain the association between noise and regulation. On the other hand, a plausible argument has been made that increased noise may be an unavoidable side-effect of regulation, i.e. due to unavoidable noise-propagation, but this is generally thought to be a detrimental effect, and does not explain our observation that selection acted to increase noise levels rather than to minimize them.

We then go on to present our general model, and explain how it clarifies these apparently contradicting observations. In particular, the model clarifies that the effects of coupling a promoter to a regulator can be usefully decomposed into a condition-response effect, whereby the mean levels of the promoter become dependent on the mean levels of the regulator, and a noise-propagation effect, whereby the noise of the promoter is increased due to noise in the regulator. The key insights that our model provides is that both of these effects can be beneficial, and that there is a continuum of possible regulatory strategies in which these two effects act in concert, varying from strategies in which all benefits come from the noise-propagation effect, to strategies where all benefits derive from the condition-response effect. In this way, our model clarifies that apparently complementary explanations such as ‘noise as a side-effect of regulation’ versus ‘noise selected as a bet hedging strategy’, are really just different parameter regimes within one unified theoretical frame work.

Finally, we return to our experimental observations, and discuss how the model helps explain them. First, we note that it would be very difficult to make quantitative predictions about what precise levels of noise would be expected for which promoters in which conditions, because this would depend on the details of how gene expression fluctuations affect fitness in different environments in the wild, which are almost impossible to know at this point. However, the model does explain the general trends observed in our Figure 2. In particular, we now discuss in some detail why our model predicts that positive correlations should be expected both between noise and the number of regulatory inputs, as well as between noise and expression plasticity.

Finally, a key assumption of our model is that elevated noise levels are, at least to some extent, the result of noise-propagation, i.e. the noise level of a promoter is determined by the noise levels of the regulators that regulate it. In the revision we now present a simple computational model that models the noise of a promoter as a linear function of the noise levels of the regulators that are known to target the promoter. We show that this model, although very crude, can explain a substantial fraction of the variation in observed noise levels. This simple model also identifies which TFs are most strongly contributing to noise levels, and we briefly discuss that the ‘noisy TFs’ that we identify are consistent with the biology of the growth conditions in which our experiments were done.

*3) More care must be taken when defining and discussing* “*transcription noise”. In the current work, neither transcription kinetics (e.g. using MS2-GFP) nor mRNA statistics (using single-molecule FISH) are directly measured, but instead a protein (GFP) reporter is used. Thus, interpreting the measured variability as* “*transcription noise” involves important caveats (see e.g. Paulsson, Nature 2004; Pedraza and Paulsson, Science 2008). These need to be explicitly discussed*.

We agree with the reviewers that we did not discuss this point in sufficient detail and have expanded the discussion of this issue in the revision. In particular, the discussion that we added in the revision covers the following points. First, we note that the reporter constructs we use were designed with the aim of measuring transcriptional noise, i.e. attempts were made to miminize the variation in the translation rates, the mRNA life-time distributions of the constructs, and the protein decay rates. Regarding the latter, since the GFP protein is long-lived, the protein decay is dominated by dilution, which, since all constructs show virtually identical growth rates, are virtually identical for all constructs.

Regarding variation in translation and mRNA decay, we note that the only difference be-tween the constructs is the promoter sequence fused upstream of GFP. As a result, the mRNAs transcribed from different constructs vary only in the few nucleotides between the TSS and the insertion site. In addition, downstream of the insertion site, is a fixed 5' UTR with a strong ribosomal binding site, intended to ensure similar translation rates of all constructs. To confirm that translation rates vary little across the constructs we performed qPCR experiments that show a high correlation between protein and mRNA levels. Since dilution/decay rates of proteins are constant, this implies that translation rates indeed vary little across the constructs, as we detail in Appendix 1. Although we did not directly measure the variation in mRNA decay rates across the constructs, the fact that the mRNAs only vary in the initial few nucleotides downstream of TSS, and the fact that these variations do not appear to significantly affect translation rates, suggest that the variations in mRNA decay rates are likely also modest.

Together these observations suggest that the observed large differences in GFP number fluctuations result mostly from transcriptional noise. This is not to say that other fluctuations do not contribute to the noise, indeed all constructs are subject to noise in translation, to fluctuations in growth rates, to fluctuations in the copy number of the plasmid, and so on. However, these all contribute the same amount of noise for each construct. Indeed, we observe a fixed minimal amount of extrinsic noise that is exhibited by all our constructs. However, the differences in the noise levels across constructs are most likely due to differences in transcriptional noise, for the reasons given above. The fact that there is a very significant association between noise level and the occurrence of binding sites for particular TFs further supports this interpretation.

*4) In multiple places, the authors make very strong statements, which require additional evidence to support them or otherwise need to be reworded. Examples*:

“*Many studies have used* circumstantial evidence *to suggest that selection generally acts to minimize noise […] there is* empirical evidence *that selection has acted to increase expression noise in some cases” (Introduction). What makes the former studies merely* “*circumstantial” while the latter rise to the level of* “*empirical*”*?*

We agree this was not clear and have added clarification as to what precisely these studies show, i.e. showing association between noise and proxies for organismal fitness (which we called ‘circumstantial’), versus showing a causal relationship between noise and growth (which we called ‘empirical’).

“*Whenever fluctuations are small relative to the mean, which applies to most promoters” (Results). No reference is given to support this statement*.

We have now clarified this. Our own results show that, for the large majority of promoters, the fluctuations are small relative to the mean (which manifests itself as the variance in log-scale being clearly less than 1, i.e. as shown in Figure 1).

“*We concluded that constitutively expressed* E. coli *promoters exhibit low excess noise levels by default” (Results). How was this conclusion reached? In particular, what is a* “*constitutive promoter” (or* “*unregulated promoter”, as mentioned later in the same section of the manuscript)? All known promoters in* E. coli *are regulated by multiple parameters: growth rate, nutritional state, temperature, etc. So how can a specific promoter be* “*constitutive”/*“*unregulated*”*?*

Although the term ‘constitutive promoter’ is commonly used in the field, we agree with the reviewers that it is often not clearly defined what precisely is meant with the term, and we are here also guilty of being vague about the term. We now realize our statement in fact confounded two separate issues that we have now tried to separate more clearly in the revision. First, and most importantly, our experiments show that promoters which were only selected to show a particular (constant) expression level in one given condition, without selecting on their noise properties, and starting from completely random sequence, naturally exhibit low noise. This is what we mean when we say that promoters have low noise ‘by default’. Second, our analysis of the noise levels of the native promoters shows that native promoters that have no known regulatory sites beyond the sigma-factor binding site of the polymerase, also tend to show low noise levels. This is what we call ‘constitutive’ promoters, i.e. not regulated by specific TFs. In the revision these points have been clarified.